# DISCO-DANCE:

## LEARNING TO DISCOVER SKILLS WITH GUIDANCE

### ABSTRACT

Unsupervised skill discovery (USD) allows agents to learn diverse and discriminable skills without access to pre-defined rewards, by maximizing the mutual information (MI) between skills and states reached by each skill. The most common problem of MI-based skill discovery is insufficient exploration, because each skill is heavily penalized when it deviates from its initial settlement. Recent works introduced an auxiliary reward to encourage the exploration of the agent via maximizing the state's epistemic uncertainty or entropy. However, we have discovered that the performance of these auxiliary rewards decreases as the environment becomes more challenging. Therefore, we introduce a new unsupervised skill discovery algorithm, skill **disco**very with gui**dance (DISCO-DANCE)**, which (1) selects the *guide skill* which has the highest potential to reach the unexplored states, (2) guide other skills to follow the *guide skill*, then (3) the guided skills are diffused to maximize their discriminability in the unexplored states. Empirically, DISCO-DANCE substantially outperforms other USD baselines on challenging environments including two navigation benchmarks and a continuous control benchmark.

## 1 INTRODUCTION

In recent years, Deep Reinforcement Learning (DRL) has shown great success in various complex tasks, ranging from playing video games (Mnih et al., 2015; Silver et al., 2016) to complex robotic manipulation (Andrychowicz et al., 2017; Gu et al., 2017). Despite their remarkable success, most DRL models focus on training from scratch for every single task, which results in significant inefficiency. In addition, the reward functions adopted for training the agents are generally handcrafted, acting as an impediment that prevents DRL to scale for various real-world tasks. For these reasons, there has been an increasing interest in training task-agnostic policies without access to a pre-defined reward function (Campos et al., 2020; Eysenbach et al., 2018; Gregor et al., 2016; Hansen et al., 2019; Laskin et al., 2021; Liu & Abbeel, 2021; Sharma et al., 2019; Strouse et al., 2022; Park et al., 2022; Laskin et al., 2022; Shafiullah & Pinto, 2022). This training paradigm falls in the category of Unsupervised Skill Discovery (USD) where the goal of the USD is to acquire diverse and discriminable behaviors, known as *skills*. These pre-trained skills can be utilized as useful primitives or directly employed to solve various downstream tasks.

Most of the previous studies in USD (Achiam et al., 2018; Eysenbach et al., 2018; Gregor et al., 2016; Hansen et al., 2019; Sharma et al., 2019) discover a set of diverse and discriminable skills by maximizing the self-supervised, intrinsic motivation as a form of reward. Commonly, mutual information (MI) between the skill's latent variables and the states reached by each skill is utilized as this self-supervised reward. However, it has been shown in recent research that solely maximizing the sum of MI rewards is insufficient to explore the state space because the agent receives larger rewards for visiting known states rather than for exploring the novel states asymptotically (Campos et al., 2020; Liu & Abbeel, 2021; Strouse et al., 2022).

To ameliorate this issue, recent studies designed an auxiliary exploration reward that incentivizes the agent when it succeeds in visiting novel states (Strouse et al., 2022; Lee et al., 2019; Liu & Abbeel, 2021). However, albeit provided these auxiliary rewards, previous approaches do not often work efficiently in complex environment. Fig. 1 conceptually illustrates how previous methods work ineffectively in a simple environment. Suppose that the upper region in Fig. 1a is hard to reach

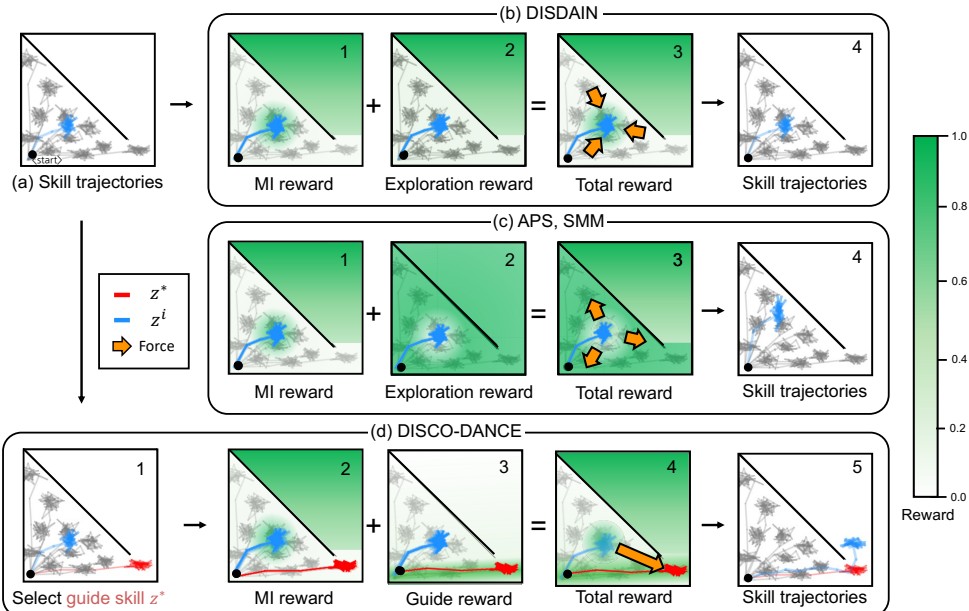

Figure 1: **Conceptual illustration of previous methods and DISCO-DANCE.** Each skill is shown with a grey-colored trajectory. Blue skill $z^i$ indicates an unconverged skill. Here, (b,c) illustrates a reward landscape of previous methods, DISDAIN, APS, and SMM. (b) DISDAIN fails to reach upper region due to the absence of a pathway to the unexplored states. (c) APS and SMM fail since they do not provide exact direction to the unexplored states. On the other hand, (d), DISCO-DANCE directly guides $z^i$ towards selected guide skill $z^*$ which has the highest potential to reach the unexplored states. Detailed explanation of the limitations of each baselines is described in Section 2.2.

with MI rewards, resulting in obtaining skills which are stuck in the lower-left region. To make these skills explore the upper region, previous methods provide auxiliary exploration reward using intrinsic motivation (e.g., disagreement, curiosity based bonus). However, since they do not indicate exactly which *direction* to explore, it becomes more inefficient in challenging environments. We detail the limitations of previous approaches in Section 2.2.

In response, we design a new exploration objective that aims to provide *direct guidance* to the unexplored states. To encourage skills to explore the unvisited states, we first pick a guide skill $z^*$ which has the highest potential to reach the unexplored states (Fig. 1(d-1)). Next, we select the skills which will move towards a guide skill (e.g., relatively unconverged skills; receiving low MI rewards). Then they are incentivized to follow the guide skill, aiming to leap over the region with low MI reward (Fig. 1(c-4)). Finally, they are diffused to maximize their distinguisability (Fig. 1(c-5)), resulting in obtaining a set of skills with high state-coverage. We call this algorithm as skill **disco**very with gui**dance (DISCO-DANCE)** and is further presented in Section 3. DISCO-DANCE can be thought of as filling the pathway to the unexplored region with a positive dense reward.

In Section 4, we demonstrate that DISCO-DANCE outperforms previous approaches with auxiliary exploration reward in terms of state space coverage and downstream task performances in two navigation environments (2D mazes and Ant mazes), which have been commonly used to validate the performance of the USD agent (Campos et al., 2020; Kamienny et al., 2021). Furthermore, we also experiment in DMC (Tunyasuvunakool et al., 2020), and show that the learned set of skills from DISCO-DANCE provides better primitives for learning general behavior (e.g., run, jump and flip) compared to previous baselines.

## 2 PRELIMINARIES

In Section 2.1, we formalize USD and explain the inherent pessimism that arise in USD. Section 2.2 describes existing exploration objectives for USD and the pitfalls of these exploration objectives.

### 2.1 Unsupervised Skill Discovery and Inherent Exploration Problem

Unsupervised Skill Discovery (USD) aims to learn *skills* that can be further utilized as useful primitives or directly used to solve various downstream tasks. We cast USD problem as discounted, finite horizon Markov decision process $\mathcal{M}$ with states $s \in \mathcal{S}$, action $a \in \mathcal{A}$, transition dynamics $p \in \mathcal{T}$, and discount factor $\gamma$. Since USD trains the RL agents to learn diverse skills in an unsupervised manner, we assume that the reward given from the environment is fixed to $0$. The skill is commonly formulated by introducing a skill latent $z \in \mathcal{Z}$ to a policy $\pi$ resulting in a latent-conditioned policy $\pi(a|s, z)$. Here, the skill's latent variable $z$ can be represented as an one-hot (i.e., discrete skill) or a continuous vector (i.e., continuous skill). In order to discover a set of diverse and discriminable skills, a standard practice is to maximize the mutual information (MI) between state and skill's latent variable $I(S; Z)$ (Achiam et al., 2018; Eysenbach et al., 2018; Gregor et al., 2016; Hansen et al., 2019; Sharma et al., 2019).

$$I(Z, S) = -H(Z|S) + H(Z) = \mathbb{E}_{z \sim p(z), s \sim \pi(z)}[\log p(z|s) - \log p(z)] \tag{1}$$

Since directly computing the posterior $p(z|s)$ is intractable, a learned parametric model $q_\phi(z|s)$, which we call *discriminator*, is introduced to derive the lower-bound of the MI instead.

$$I(Z, S) \geq \tilde{I}(Z, S) = \mathbb{E}_{z \sim p(z), s \sim \pi(z)}[\log q_\phi(z|s) - \log p(z)] \tag{2}$$

Then, the lower bound is maximized by optimizing the skill policy $\pi(a|s, z)$ via any RL algorithm with reward $\log q_\phi(z|s) - \log p(z)$ (referred to as $r_{\text{skill}}$). Note that each skill-conditioned policy gets different reward for visiting the same state (i.e., $r_{\text{skill}}(z_i, s) \neq r_{\text{skill}}(z_j, s)$). It results in learning skills which visit different states, making them discriminable.

However, maximizing the MI objective is insufficient to fully explore the environment due to an inherent pessimism of its objective (Campos et al., 2020; Liu & Abbeel, 2021; Strouse et al., 2022). When the discriminator $q_\phi(z|s)$ succeeds to distinguish the skills, the agent receives larger rewards for visiting known states rather than for exploring the novel states. This lowers the state coverage of a given environment, suggesting that there are limitations in discovering what skills are available (i.e., achieving a set of skills that only reach in a limited state space).

Therefore, recent studies in USD provide auxiliary rewards to overcome pessimistic exploration.

### 2.2 Previous Exploration Bonus and Its Limitations

DISDAIN (Strouse et al., 2022) trains an ensemble of $N$ discriminators and rewards the agent for their disagreement, represented as $H(\frac{1}{N} \sum_{i=1}^{N} q_{\phi_i}(Z|s)) - \frac{1}{N} \sum_{i=1}^{N} H(q_{\phi_i}(Z|s))$. Since states which have not been visited frequently will have high disagreement among discriminators, DISDAIN implicitly encourages the agent to move to novel states. However, since such exploration bonus is a *consumable* resource that diminishes as training progresses, most skills will not benefit from this if other skills reach these new states first and earn the bonus reward.

We illustrate this problem in Fig. 1b. Suppose that all skills remain in the lower left states, which are easy to reach with MI rewards. Since the states in lower left region is accumulated in the replay buffer, disagreement between discriminators remains low (e.g., low exploration reward in lower left region). Therefore, there will be no exploration reward left in these states. This impedes $z_i$ from escaping its current state to unexplored states, as shown in Fig. 1b-4.

On the other hand, SMM encourages the agent to visit the states where it has not been before using learned density model $d_\theta$ (Lee et al., 2019). APS incentivizes the agent to maximize the marginal state entropy via maximizing the distance of the encoded states $f_\theta(s_t)$ between its k nearest neighbor $f_\theta^k(s_t)$ (Liu & Abbeel, 2021).

$$\begin{aligned} r_{\text{exploration}}^{\text{SMM}} &\propto -\log d_\theta(s) \\ r_{\text{exploration}}^{\text{APS}} &\propto \log ||f_\theta(s) - f_\theta^k(s)|| \end{aligned} \tag{3}$$

These rewards push each skill out of its converged states (in Fig. 1c). However, they still do not provide a *specific direction* on where to move in order to reach unexplored states. Therefore, in a difficult environment with a larger state space, it is known that these exploration rewards can

operate inefficiently (Campos et al., 2020; Ecoffet et al., 2019). In the next section, we introduce a new exploration objective for USD which addresses these limitations and outperforms prior methods on challenging environments.

## 3 METHOD

Unlike previous approaches, we design a new exploration objective where the *guide skill* $z^*$ directly influences other skills to reach explored regions. DISCO-DANCE consists of two stages: (i) selecting guide skill $z^*$ and the *apprentice skills* which will be guided by $z^*$, and (ii) providing guidance to apprentice skills via *guide reward*, which will be described in Section 3.1 and 3.2, respectively.

### 3.1 SELECTING GUIDE SKILL AND APPRENTICE SKILLS

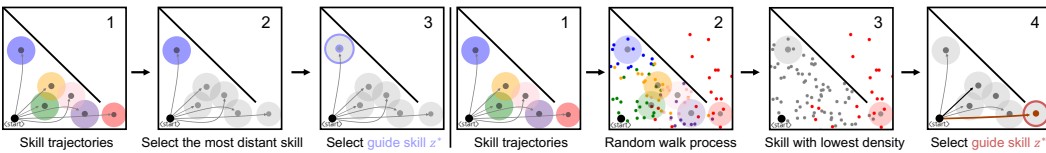

Figure 2: **Guide skill selection.** An illustration of a random walk process which finds the skill with a highest potential that can reach an unexplored states.

**Guide skill.** We recall that our main objective is to obtain a set of skills that provides high state space coverage. To make other skills reach unexplored state, we define the *guide skill* $z^*$ as the skill which is most *adjacent* to the unexplored states. One naive approach in selecting the guide skill is to choose the skill whose terminal state is most distant from the other skills' terminal state (e.g., blue skill in Fig. 2.1). However, such selection process does not take into account whether the guide skill's terminal state is neighboring promising unexplored states. In order to approximate whether a skill's terminal state is near potentially unexplored states, we utilize a random walk process (Fig. 2). To be specific, when there are $P$ skills, (i) we first perform $R$ random walks from each skill's terminal state and collect a total of $PR$ number of random walk arrival states. Repeat (i) process for $M$ times, collect $PRM$ number of states in total. (ii) Then we pinpoint the state in the lowest density region among the collected random walk arrival states and select the skill which that state originated from as the guide skill $z^*$. For (ii), one could use any algorithm to measure the density of the random walk state distribution. For our experiments, we utilize simple k nearest neighbor algorithm,

$$z^* := \underset{p \in \{1,\dots,P\}}{\arg\max} \; \underset{r \in \{1,\dots,RM\}}{\max} \frac{1}{k} \sum_{s_{pr}^j \in N_k(s_{pr})} ||s_{pr} - s_{pr}^j||_2$$

$$\text{where } s_{pr} = r\text{-th random walk arrival state of skill } p$$
$$N_k(\cdot) = \text{k-nearest neighbors} \tag{4}$$

In practice, in environment with long-horizon such as DMC, our random walk process may cause sample inefficiencies. Therefore, we present an efficient random walk process which we describe in detail in Appendix E. Further ablation experiments on how to select guide skills are available in Section 4.4.1.

**Apprentice skills.** We select *apprentice skills* as the skills with low discriminability (i.e., skill $z^i$ with low $q_\phi(z^i|s)$; which fails to converge) and move them towards the guide skill. If most of the skills are already well discriminable (i.e., high MI rewards), we simply add new skills and make them as apprentice skills. This will leave converged skills intact and send new skills to unexplored states. Since the total number of skills is gradually increasing as the new skills are added, this would bring the side benefit of not relying on pre-defined number of total skills as a hyperparameter.

We note that adding new skills during training is generally difficult to apply to previous algorithms, because the new skills will also face the pessimistic exploration problem. The new skills simply

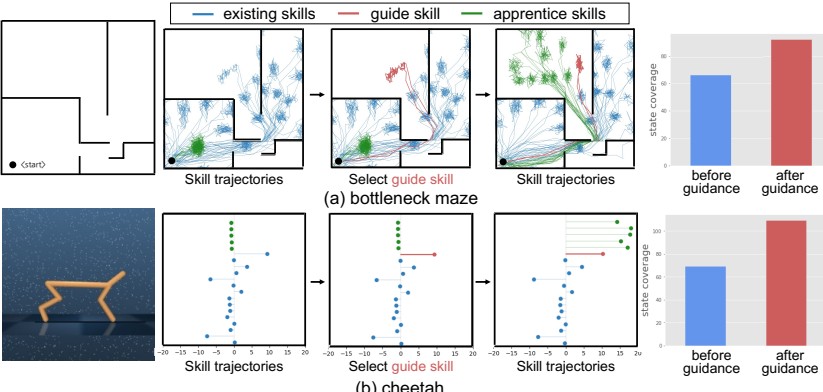

Figure 3: **Qualitative results of the guiding procedure** in (a) navigation and (b) continuous control.

converge by overlapping with existing skills (e.g., left lower room in Fig 3), which exacerbates the situation (i.e., reducing discriminator accuracy without increasing the state coverage). In Appendix D, we empirically show that this curriculum procedure brings benefit to our methods, while the improvement over the baselines are marginal.

### 3.2 PROVIDING DIRECT GUIDANCE VIA AUXILIARY REWARD

After selecting the guide skill $z^*$ and apprentice skills, we now formalize the exploration objective, considering two different aspects: (i) objective of the guidance, and (ii) the degree of the guidance. It is crucial to account for these desiderata since strong guidance will lead apprentice skills to simply imitate the guide skill $z^*$ whereas the weak guidance will not be enough for skills to overcome pessimistic exploration.

In response, we propose a exploration objective that enables our agent to learn with guidance. We integrate these considerations into a single soft constraint as

$$
\begin{aligned}
\underset{\theta}{\text{maximize}} \; & E_{z^i \sim p(z), s \sim \pi_\theta(z^i)} \big[ r_{\text{skill}} - r_{\text{guide}} \big] \\
\text{where} \; & r_{\text{skill}} = \log q_\phi(z^i|s) - \log p(z^i), \\
& r_{\text{guide}} = -\alpha \, \mathbb{I}\big(q_\phi(z^i|s) \le \epsilon\big) \left(1 - q_\phi(z^i|s)\right) D_{\text{KL}}(\pi_\theta(a|s, z^i) || \pi_\theta(a|s, z^*)).
\end{aligned}
\tag{5}
$$

As we describe in Section 3.1, we select skills with low discriminator accuracy (i.e., $\mathbb{I}(q_\phi(z^i|s) \le \epsilon)$) as apprentice skills. For (i), we minimize the KL-divergence between the apprentice skill and the guide skill policy (i.e., $D_{\text{KL}}(\pi_\theta(a|s, z^i) || \pi_\theta(a|s, z^*))$). For (ii), the extent to which the apprentice skills are encouraged to follow $z^*$, can be represented as the weight to the KL-divergence. We set the weight as $1 - q_\phi(z^i|s)$, to make the skills with low discriminator accuracy to be guided more.

Fig. 3 shows that this guidance boosts skill learning for both navigation environment and continuous control environment. In navigation task (Fig. 3a), the skill in the upper left is selected as a guide through the random walk process and the apprentice skills stuck in the lower left are directly dragged into the unexplored states by guide skill. In addition, in a continuous control environment (Fig. 3b), the skill that learned to *run* is selected as the guide skill, leading the apprentice skills that are barely moving. Then the apprentice skills learns to run, in an instant. This salient observation suggests that the concept of guidance can be utilized, even for non-navigation tasks. In Section 4.4.2, we show that all three component of $r_{\text{guide}}$ are necessary through additional experiments.

## 4 EXPERIMENTS

### 4.1 EXPERIMENTAL SETUP

**Environments** We evaluate DISCO-DANCE on three different types of environments: 2D navigation, Ant Maze, and Deepmind Control Sutie (DMC). Fig 4.(a) illustrates five different layouts in

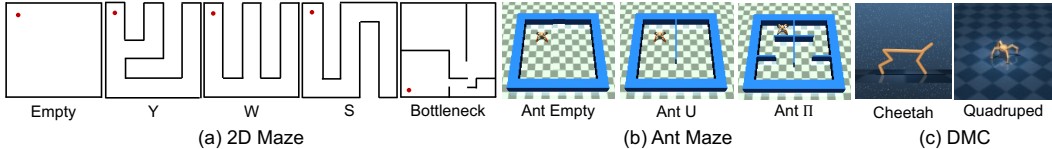

Figure 4: **Three environments to evaluate skill discovery algorithms.** (a) Fully-continuous 2D mazes with various layouts, (b) High dimensional ant navigation tasks and (c) Continuous control environment with diverse downstream tasks.

2D navigation environment. The layout becomes more challenging from empty maze to bottleneck maze (i.e., Easy-Empty, Normal-Y, W, S, Hard-Bottleneck). These environment has been commonly used for testing the exploration ability of skill-learning agents (Campos et al., 2020; Kamienny et al., 2021). The agent determines where it should move given the current x, y coordinates (2-dimensional state, action space). Fig 4.(b) shows three different layouts of AntMaze environment from (Chane-Sane et al., 2021; Nasiriany et al., 2019) which aims to evaluate the effectiveness of algorithm with high-dimensional input. In Antmaze, the state space consists of joint angles, joint velocity and the center of mass and the action space consists of torques of each joint. Following previous work (Eysenbach et al., 2018; Sharma et al., 2019), we restrain the input of the discriminator as x,y coordinates since the main goal in navigation environments is to learn a set of skills which are capable to cover the majority of the state space. Fig 4.(c) displays two different environment in DMC, Cheetah and Quadruped. DMC is selected to evaluate the algorithm's capability of learning general skills. Following URLB (Laskin et al., 2021), we consider eight different tasks (e.g., walk, run, jump, flip).

**Evaluation** We use the state space coverage and downstream task performances as our main metrics. To measure the *state coverage*, we discretize the x and y axes of the environment into 10 buckets (i.e., total 100 buckets) and count the number of buckets reached by learned skills (Fig. 4.a,b). For 2D mazes, we trained each algorithm with 2M training steps for easy and medium levels, 5M training steps for hard level (Fig. 4a). For Ant mazes, we trained for 5M training steps for the high-dimensional Ant navigation tasks (Fig. 4b). For DMC, after 2M steps of pretraining with USD algorithms, we first select the skill with maximum downstream task reward, and then finetune 100k steps for each task. (Fig. 4c). For detailed explanation, see Appendix B.1. Bold scores in Table indicate the best model performance and underlined scores indicate the second best.

**Baselines** We compare DISCO-DANCE with DIAYN (Eysenbach et al., 2018), which is the most widely used baseline in USD. We also compare DISCO-DANCE with SMM (Lee et al., 2019), APS (Liu & Abbeel, 2021) and DISDAIN (Strouse et al., 2022). We note that all baselines and DISCO-DANCE utilize discrete skills, except APS which utilizes continuous skills. In downstream tasks, we compare baselines with *random init* baseline that is initialized from scratch. We use SAC as the backbone RL algorithm (Haarnoja et al., 2018). We train DIAYN, APS and SMM using the code and hyperparameters provided by URLB (Laskin et al., 2021). In addition, we re-implement DIS-DAIN by strictly following the details of the paper. Additional details are available in Appendix B.

## 4.2 NAVIGATION ENVIRONMENTS

### 4.2.1 2D MAZES

Table 1: **State space coverages of DISCO-DANCE and baselines on 2D maze.** The results are averaged over 5 random seeds accompanied with a standard deviation.

| Models | Empty | Y | W | S | Bottleneck |
|---|---|---|---|---|---|
| DIAYN | **100.00**±0.00 | 71.20 ±5.07 | 71.40 ±4.62 | 53.80±7.01 | 53.00±4.36 |
| SMM | **100.00**±0.00 | 89.60±6.50 | 74.20±6.22 | 57.20±7.46 | 57.20±6.10 |
| APS | 95.80±4.55 | 90.20 ±5.02 | 74.20±10.62 | 77.60 ±7.50 | 61.80± 15.06 |
| DISDAIN | **100.00** ±0.00 | 82.40 ±3.91 | 84.80±11.30 | 61.80±7.33 | 60.60±2.51 |
| DISCO-DANCE | **100.00**±0.00 | **99.00**±1.41 | **94.80**±3.49 | **83.00**±1.22 | **87.40**±11.59 |

Table 1 summarizes the performance of skill learning algorithms on the 2D maze environments. Through our empirical results, we observe the following results: methods that utilize an auxiliary

reward (i.e., SMM, APS, and DISDAIN) outperform DIAYN for every maze, showing that the auxiliary exploration rewards encourage the agent to visit unseen states. However, we find out that previous studies provide less benefit on the performance as the layout becomes complicated. As we mentioned in section 2.2, we empirically discover that auxiliary exploration reward approaches are less useful when the environments getting more complicated. In contrast, for each 2D maze environment, DISCO-DANCE has negligible performance reduction with increased layout complexity. In addition, DISCO-DANCE outperforms previous methods in every 2D maze environment, exhibiting the largest performance gap against baselines in the most complex environment, bottleneck maze.

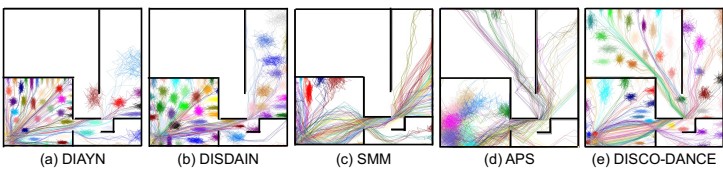

(a) DIAYN     (b) DISDAIN     (c) SMM     (d) APS     (e) DISCO-DANCE

Figure 5: **Visualization of the learned skills on bottleneck maze.** Multiple rollouts by each skill discovery algorithm.

Fig. 5 illustrates multiple rollouts of various skills learned in the 2D bottleneck maze. For the 2D bottleneck maze, the upper left room is the most difficult region to reach since it the agents need to pass multiple narrow pathways. While other baselines methods are not able to effectively explore the upper left region, only DISCO-DANCE is able to fully cover the environment.

### 4.2.2 ANT MAZES

Table 2 reports the state coverage of DISCO-DANCE and baselines. We find that among baselines, only SMM gets better performance than DIAYN. We speculate that additional reward signal may interrupt the agent to converge in high-dimensional environment. Empirically, DISCO-DANCE shows superior performance to the baselines where it achieves the best state-coverage performance in high-dimensional environments which contains obstacles (i.e., Ant

Table 2: **State space coverages of DISCO-DANCE and baselines on Ant Maze.** The results are averaged over 5 random seeds.

| Models | Ant Empty-maze | Ant U-maze | Ant Π-maze |
|---|---|---|---|
| DIAYN | 74.80±15.74 | 50.40±10.50 | 22.80±4.55 |
| SMM | **99.80±0.45** | 63.00±7.58 | 27.20±5.93 |
| APS | 73.40±33.49 | 61.20±3.96 | 27.4±8.79 |
| DISDAIN | 54.40±33.06 | 35.60±28.50 | 19.40±11.19 |
| DISCO-DANCE | 96.40±0.89 | **72.00±8.12** | **34.60±5.08** |

U-maze and Π-maze) and gets competitive results against SMM with a marginal performance gap in the environment without any obstacle (i.e., Ant Empty-maze).

In addition, we conduct an experiment to evaluate whether the learned skills could be a good starting point for downstream task which aims to reach goals (Fig. 6). We set the farthest region from the initial state as the goal state (the most difficult configurations) following Chane-Sane et al. (2021) and measure the success rate of reaching the established goal state. For methods with discrete skills, we select the skill with the maximum return (i.e., skill whose state space is closest to the goal state) and initialize the policy and value network with the selected

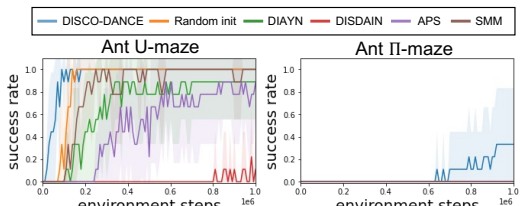

Figure 6: **Accelerating learning with policy initialization.** Curves are averaged over 9 random seeds with a standard deviation.

skill. We then fine-tune the initialized agent to maximize the reward (i.e., minimize distance to goal state) with 1M training steps. For methods with continuous skills, we follow the fine-tuning procedure in APS (Liu & Abbeel, 2021).

In Ant U-maze, DISCO-DANCE outperforms prior methods including *random init* in terms of sample efficiency, and successfully reaches the goal-state with only 100k interactions. In the most challenging environment, Ant Π-maze, only DISCO-DANCE was able to succeed in reaching the goal state within 1M environment interactions. We also observe that baselines underperform in comparison with *random init*. As shown in URLB on continuous control tasks (Laskin et al., 2021),

we conjecture that task-specific training might be more efficient than transferring general behavior. Additional details about the finetuning tasks are available in Appendix B.1.

## 4.3  DEEPMIND CONTROL SUITE

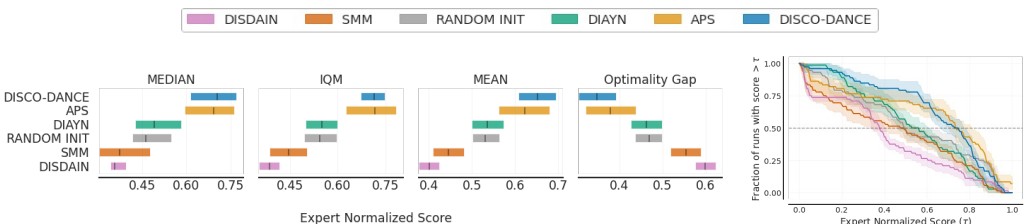

Figure 7: **Performance comparisons on Deepmind Control Suite.** Aggregate statistics and performance profiles with 95% bootstrap confidence interval are provided (Agarwal et al., 2021), which are calculated across 72 seeds (3 pretrained models × 3 seeds × 8 tasks) for all algorithms.

To demonstrate that DISCO-DANCE can also effectively learn the *general skills* in environments other than navigation related environments, we conduct additional experiments on Deepmind Control Suite (DMC) (Tunyasuvunakool et al., 2020). Fig. 7 shows the performance comparisons on DMC across 8 tasks with 9 seeds. As shown in Fig. 3b, we find that the concept of *guidance* boosts the skill learning process in the continuous control environment, achieving comparably good results in comparison with other baselines. Note that the performance of baselines with discrete skills are equal or less than the randomly initialized SAC, similarly to Ant mazes (Fig. 6). We suspect that the current finetuning strategy widely used in USD research for discrete skills (only finetune one selected skill among all skills) is not able to properly evaluate the performance of the obtained set of skills on downstream tasks. We believe that new finetuning methods that can fully leverage the characteristics of skills learned through USD will be an exciting direction for future work. We provide the full performance table at Section F.

## 4.4  ABLATION STUDIES

To further understand the role of each model components we perform two ablation studies: ablation of (1) the guide skill selection and (2) the importance of the KL coefficient in Eq. 5.

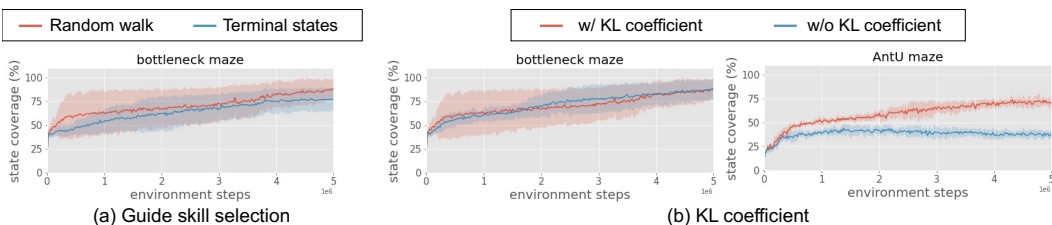

Figure 8: (a) Comparing state coverage with random walk guide skill selection and terminal state guide skill selection process. (b) Importance of KL coefficient in Eq.5.

### 4.4.1  GUIDE SKILL SELECTION PROCESS

We empirically compare our random walk guide skill selection process with the most distant terminal state guide skill selection process, in 2D bottleneck maze and AntU maze. Results in Fig. 8a shows that the random walk guide selection process is superior to the terminal state guide selection process in terms of state coverage. We also provide qualitative results where the terminal state guide selection process fails to select the skill which is closest to the unexplored region as the guide skill while the random walk guide selection process successfully finds the appropriate guide skill in Fig. 9.

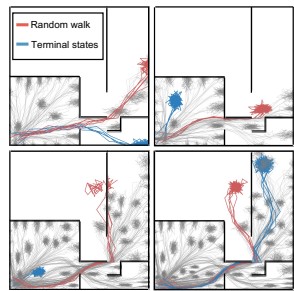

Figure 9: Qualitative results for different guide skill selection processes.

### 4.4.2 KL COEFFICIENT

We further conduct an ablation study to show the importance of the KL coefficient in the DISCO-DANCE objective. Specifically, we report the *state coverage* performance of DISCO-DANCE when trained without the KL coefficient, $\mathbb{I}\big(q_\phi(z^i|s) < \epsilon\big) \cdot (1 - q_\phi(z^i|s))$ in Equation 5 (i.e., KL coefficient as 1), to figure out the influence of guiding only apprentice skills (i.e.,g make all other skills as apprentice skills). Fig 8b shows that guiding too many skills could degrade performance in high-dimension environment and that it is important to only select unconverged skills as apprentice skills for effective learning.

## 5 RELATED WORK

**Previous research in Unsupervised Skill Discovery.** Several approaches have been made to learn a set of useful skills in fully-unsupervised manner. Since MI based method suffers from inherent pessimism, EDL (Campos et al., 2020) decomposes training procedure, first explores the environment with SMM (Lee et al., 2019), and then utilize a VAE to discover skills. Since they completely separate between the exploration phase and skill learning phase, EDL relies heavily on the performance of exploration module. CIC (Laskin et al., 2022) uses noise contrastive estimation (NCE) (Gutmann & Hyvärinen, 2010) to enable learning high-dimensional skills for diverse behaviors. LSD (Park et al., 2022) utilizes Lipschitz-constrained regularization term to learn dynamic behaviors, and has shown that it works well in downstream task in a zero-shot manner. DISk (Shafiullah & Pinto, 2022) aims to learn diverse skills which can quickly adapt to a non-stationary environment. To achieve this, Disk learns skills in an incremental fashion, which encourages a new skill to be consistent within itself and to be diverse from the previous skills. UPSIDE (Kamienny et al., 2021) utilizes a hierarchical tree-structure which execute multiple skills sequentially to reach distant states. Further discussion of the differences between DISCO-DANCE and UPSIDE are available in Appendix G.

**Guidance based exploration in RL.** Go-Explore (Ecoffet et al., 2021; 2019) stores the visited states in the buffer and starts re-exploration from the samples states in the buffer to boost the exploration process. DISCO-DANCE and Go-Explore share a similar motivation that the agent guides itself: DISCO-DANCE learns from other skills and Go-Explore learns from previous experiments that have reached a novel states. Another line of guided exploration is to utilize a KL-regularization between the policy and the demonstration (i.e., guide) (Chane-Sane et al., 2021; Pertsch et al., 2020). SPIRL (Pertsch et al., 2020) which is an offline supervised skill discovery method, denotes action sequences as *skill*($z$) and learns the *skill* prior $p(z|s)$ using offline data. For finetuning, it reduces the KL divergence between the agent's policy and the skill prior as an auxiliary reward to accelerate exploration for the new task. RIS (Chane-Sane et al., 2021) has been proposed to stimulate efficient exploration for goal-conditioned RL, where reaching a distant goal is challenging due to the sparse rewards. RIS efficiently learns to reach distant goals by imitating the policy that aims to reach a subgoal generated between its position and distant goal, which makes the rewards more accessible.

## 6 CONCLUSION

In this paper, we introduce DISCO-DANCE, a novel, efficient exploration strategy for USD. It directly guides the skills towards the unexplored states, by forcing them to follow the *guide skill*. We provide quantitative and qualitative experimental results which demonstrates that DISCO-DANCE outperforms existing methods in two navigation benchmarks and a continuous control benchmark.

Although not thoroughly studied in this paper, we think that there is still enough room for improvement in finetuning strategies (Laskin et al., 2021). Since USD learns many different task-agnostic behaviors, a new fine-tuning strategy that can take advantage of these points would make the downstream task performances more powerful. Also, while this paper focuses on learning discrete skills, the concept of our *guide and explore* can be extended to continuous skill spaces. DISCO-DANCE can easily be extended to jointly work with other algorithms which aim to learn diverse behavioral patterns (Achiam et al., 2018). Since DISCO-DANCE accelerates the exploration process, such algorithms can benefit by learning skills from a wider reachable state-space. Finally, discussion of limitation and future directions are available in Appendix H.

**Reproducibility Statement** To ensure reproducibility, we provide a (i) pseudo-code and implementation detail in Appendix A, (ii) all hyperparameters in Table 3, (iii) thorough explanation of three environments in Appendix B.1, and (iv) implementation details of other baselines in Appendix B.2. We will share the self-contained code when the discussion period opens.

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

# A   DETAILS OF DISCO-DANCE

---
**Algorithm 1:** Skill Discovery through Guidance

---
1  Initialize replay buffer $D$, skills $z^1, ..., z^n$, KL coefficient $\alpha$, discriminator accuracy cutoff $\epsilon$
2  Initialize policy $\pi_\theta(a|s, z)$, Skill discriminator $q_\phi(z|s)$
3  **for** $k = 1, 2, ...$ **do**
4      Collect experience in $D$ using $\pi_\theta$ in the environment
5      Sample batch $(s_t, a_t, s_{t+1}, z) \sim D$
6      **if** *Most existing skills are discriminable enough* **then**
7          Select guide skill $z^* \leftarrow$ *find_guide_skill($\pi_\theta, z^1, ..., z^n$)*
8      $r_{\text{skill}} = \log q_\phi(z|s_{t+1}) - \log p(z)$
9      $r_{\text{guide}} = -\mathbb{I}\big(q_\phi(z^i|s) < \epsilon\big) \cdot (1 - q_\phi(z|s)) \cdot D_{\text{KL}}(\pi_\theta(a_t|s_t, z)||\pi_\theta(a_t|s_t, z^*))$
10     $r = r_{\text{skill}} + \alpha \cdot r_{\text{guide}}$
11     Update $\pi_\theta$ to maximize sum of $r$
12     Update $q_\phi$ to maximize $\log q_\phi(z|s_t)$

---

The full DISCO-DANCE algorithm is shown in Algorithm 1. We list the implementation details of below.

- Entropy term in SAC Haarnoja et al. (2018) can be seen as KL divergence between the policy and uniform distribution with a constant Pertsch et al. (2020) $(\mathcal{H}(\pi(\cdot|s_t, z_t)) \propto -D_{KL}(\pi(\cdot|s_t, z_t)||U(\cdot))$. Instead of replacing the entropy term with guide term in DISCO-DANCE, we found that using both term simultaneously helped stabilize learning by enjoying the advantage of maximum entropy RL.

- We utilize a KL-regularization between the overlapping skill and the guide skill policy. However, the guide skill policy changes during training. To improve the stability of our method, we use a separate target policy network for the guide skill. More precisely, when we select the guide skill, we clone the policy network to obtain a target policy network and use the target network for generating the fixed guide skill policy outputs.

- For adding new skills, we need to define hyperparameters: when to extend skills and how many to increase. We gradually extend skills as the discriminator converges. In detail, count the number of skills that have low discriminator accuracy (i.e., less than $\epsilon$) and if the number is less than the threshold (i.e., $\rho$), extend $\rho$ skills.

- As the number of skills is gradually increasing, the discriminator struggles to learn due to the sparse update for each skill by the decreased probability of sampling each skill. We mitigate this issue by resorting a skill sampling scheme where each skill sampling weight is proportionate to its discriminator error (e.g., skills with high discriminator error are more likely to be sampled

- When extending new skills, we initialize their policy weights with the guide skill policy weights rather than random weights as in SPIRL Pertsch et al. (2020). This accelerates exploration by encouraging the newly added skills to directly reach the unexplored regions.

- After following the guide skill, the overlapped skills need to get diffused to maximize their distinguishability. When getting diffused, KL divergence between the overlapped skills and the guide skill becomes larger. To allow this, the skills get the same KL divergence when the KL divergence is lower than a threshold (i.e., $\eta$).

- We include the number of timesteps used in guide skill selection in the total pretraining timesteps. We measure the state coverage with the checkpoints of each pretrained model with earlier timesteps. For example, in 2D bottleneck maze, $135,000$ samples are used for selecting guide skill, therefore we report the state coverage with model checkpoints at $t = 4,500,000$ (<5M - 135,000).

- For Antmaze, we select 3 pretrained models among 5 random seeds and fine-tuned each model for 3 random seeds. In Fig 6, All curves are averaged over 9 random seeds (3 pretrained models $\times$ 3 seeds for all algorithms.

# B  FURTHER DETAILS ON ENVIRONMENTS AND HYPERPARAMETERS

## B.1  ENVIRONMENTS AND EVALUATION

**2D maze** The width and height of easy and medium level mazes are both 5, and 10 for hard level maze Campos et al. (2020); Kamienny et al. (2021). There is no explicit terminal state, and episode ends only when the maximum timestep (30 for easy and medium levels and 50 for hard level) is reached.

**Ant maze** The width and height of U maze are (7,7) and Empty, Π-shaped maze are (8,8) respectively Chane-Sane et al. (2021); Nasiriany et al. (2019). Similar to 2d-maze, episode ends only when the maximum timestep (400 for every mazes) is reached. For goal-reaching downstream tasks, we design a distance between current state and goal state as a penalizing reward (i.e., $-1 \times$ distance $=$ reward). In detail, directly computing distance with x,y coordinates might be inaccurate because there can be obstacles (i.e., walls) between two states. Therefore, we design a reward function that returns the actual distance that goes around when there is an obstacle between two states. Then, we normalize the reward to set the lowest value as $-1$.

**Deepmind Control Suite** We conduct our environment in a total of eight downstream tasks provided by URLB (Laskin et al., 2021). In detail, we utilize four tasks each in two continuous control domains: Run, Run backward, Flip, and Flip backward for Cheetah and Jump, Run, Stand, and Walk for Quadruped. Following the convention of Laskin et al. (2021), we define expert score as the performance of randomly initialized SAC for 2M timestep, whereas only 100k timesteps for finetuning are allowed. For DIAYN, SMM, DISDAIN and DISCO-DANCE (discrete skills), we finetune the skill with maximum downstream task reward (same as Ant mazes). For APS (continuous skill), we first randomly select an arbitrary skill z and rollout episodes, then solve thde linear regression problem to select the task-specific skill z (following same protocol in APS paper). After the skill is selected for finetuning, for all algorithms, selected skill is fine-tuned for 100K steps. Note that this is different from URLB in that the skill selection process is not included in 100K, but this is still a fair comparison as all algorithms use the same number of interactions (samples) to select skills.

## B.2  HYPERPARAMETERS

Detailed hyperparameters of our method DISCO-DANCE are listed in the table 3.

Table 3: **Hyperparameters used in DISCO-DANCE.**

| Hyperparameter | 2D maze | Ant maze | DMC |
|---|---|---|---|
| Hidden dim | 128 | 1024 | 1024 |
| Batch size | 64 | 512 | 1024 |
| Skill trajectory length | easy, normal: 30 hard: 50 | 400 | 1000 |
| Initial number of skill | 30 | 10 | 10 |
| Replay buffer size | $10^6$ | $10^6$ | $10^6$ |
| Optimizer | Adam | Adam | Adam |
| Discriminator learning rate | $3 \times 10^{-4}$ | $3 \times 10^{-4}$ | $3 \times 10^{-4}$ |
| Critic learning rate | $3 \times 10^{-4}$ | $3 \times 10^{-4}$ | $3 \times 10^{-4}$ |
| Actor learning rate | $3 \times 10^{-4}$ | $3 \times 10^{-4}$ | $3 \times 10^{-4}$ |
| RL algorithm | Soft Actor Critic | Soft Actor Critic | Soft Actor Critic |
| discount $\gamma$ | 0.99 | 0.99 | 0.99 |
| Entropy coef ($\beta$) | 0.2 | 0.6 | 0.2 |
| Select guide | K nearest neighbor | K nearest neighbor | K nearest neighbor |
| KL coef ($\alpha$) | $10^{-4}$ | $10^{-4}$ | $10^{-4}$ |
| Number of skills to extend | 5 | 5 | 5 |
| Extending threshold $\rho$ | 5 | 5 | 5 |
| Accuracy threshold $\epsilon$ | 0.5 | 0.5 | 0.5 |
| KL threshlod ($\eta$) | 10 | 10 | 10 |

We construct a two-layer MLP with 128 (2D maze) or 1024 (Ant maze, DMC) hidden units, and use adam Kingma & Ba (2014) optimizer with $3 \times 10^{-4}$ learning rate. For each skill, we sample 50 (bottleneck maze), 400 (Ant maze), 1000 (DMC) length trajectory and train the agent with SAC Haarnoja et al. (2018) using 64 (2D maze), 512 (Ant maze), 1024 (DMC) batch size. We set the discount factor $\gamma$ as 0.99 and the SAC entropy coefficient $\beta$ as 0.2 for 2D maze and DMC, 0.6 for Ant maze.

For DISDAIN, we search the number of ensembles from [5, 10, 20, 40] and disdain reward coefficient from [10, 50, 100]. We train DISDAIN using 40 size of ensembles and set DISDAIN reward weight as 100, for all environments. We also train separate Q-funcetions for $r_{skill}$ and $r_{disdain}$ and add two Q-values with weight 0.7 (reward = $r_{skill} + 0.7 \, r_{disdain}$) following (Strouse et al., 2022) .

For APS, we search the value of k (i.e., number of neighbors in k nearest neighbor algorithm) from [5, 10] and the dimension of skills from [5, 10]. For 2d-mazes, we utilize 10 for k and 5 for skill dimensions. For Ant mazes, we utilize 5 for k and 5 for skill dimensions. For DMC, we utilize 12 for k and 10 for skill dimensions.

For SMM, we strictly follow the implementation detail from the paper. We search the entropy coefficient from [0.0005, 0.005, 0.01, 0.1, 0.25, 0.5, 1]. We utilize entropy coefficient as 0.1 for 2D maze and Antmaze, 0.01 for DMC.

For the methods using the curriculum approach, we use the initial number of skills 30 for the 2D maze, and 10 for the Antmaze and DMC. During training, count the number of skills in which discriminator accuracy is less than the threshold $\epsilon = 0.5$, and if it is less than the extending threshold $\rho = 5$, extend 5 number of skills.

## C    ADDITIONAL ABLATION STUDIES

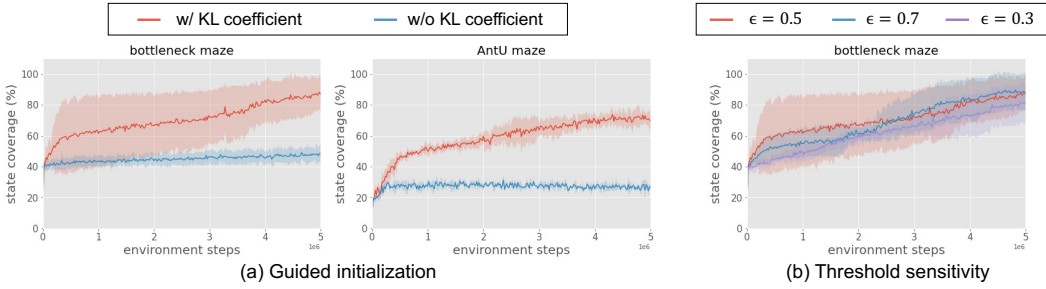

(a) Guided initialization                    (b) Threshold sensitivity

Figure 10: **Ablation studies.** (a) Guided initialization. (b) Sensitivity to the accuracy threshold.

During the curriculum learning procedure, we initialize the weights of newly added skills using the guide skill to directly reach the unexplored states. To analyze the importance of initialization, we compare this approach to an ablation that training from scratch. In Fig 10(a), initializing the weights by using the guide skill makes the performance better. We speculate that guide initialization is directly minimizing the kl divergence with guide skill, it makes the agent start with the maximum guide reward. In order to further analyze the sensitivity of the accuracy threshold $\epsilon$, we conduct an ablation study. We vary the accuracy threshold $\epsilon$ from $\{0.3, 0.5, 0.7\}$ and fix the remaining hyperparameters. In Fig 10(b), we observe that the performance of DISCO-DANCE is robust to the accuracy threshold $\epsilon$.

## D    COMPARING CURRICULUM APPROACH EFFECTS ON USD METHODS

To demonstrate DISCO-DANCE is highly compatible with increasing the number or skills during training, we also combine the curriculum approach to the baselines which utilize discrete skill spaces

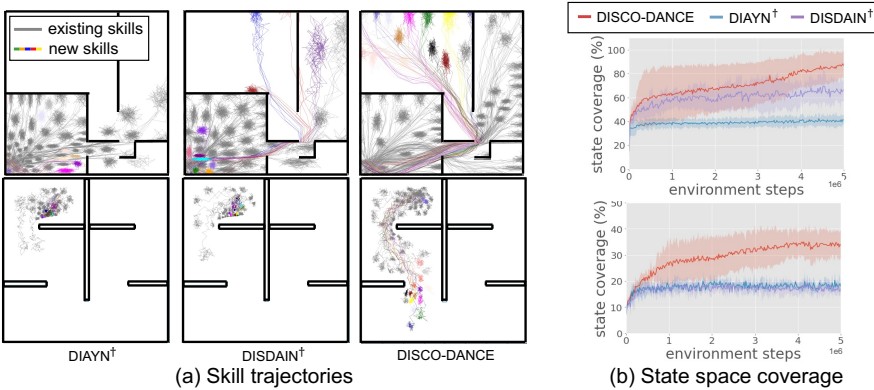

Figure 11: **Gradually increasing the number of skills for DISCO-DANCE, DIAYN, and DIS-DAIN.** (a) Qualitative results on bottleneck maze and Ant Π-maze. Colored skills indicate new skills and grey skills are previously converged skills. (b) Training curves on each environment.

for comparison. In order to empirically verify whether previous approaches can also benefit from curriculum procedure, we combined the curriculum approach mentioned in Section 3.1 to DIAYN and DISDAIN . As shown in Fig. 11a, we discover that DIAYN and DISDAIN are not able to benefit from curriculum learning and that the newly added skills are mainly rooted near the initial starting point. This results in making the covered state space more crowded, so the curriculum approach not only does not help to cover more state spaces, but exacerbate the situation by reducing discriminator accuracy without increasing the state coverage. However for DISCO-DANCE, the newly added skills are able to directly reach unexplored regions with the aid of the guide skill.

## E  EFFICIENT RANDOM WALK PROCESS

Our random walk process in Section 3.1 is as follows

1. After performing rollout for all $P$ learned skills till the terminal state, we perform $R$ random walks and collect $P * R$ number of random arrival states ($R$ is a hyper-parameter and we set $R$ as 0.2 * time horizon $T$ for all environments). Repeat this process a total of $M$ times.
2. Pinpoint the state in the lowest density region among the $P * R * M$ number of states. We measure the density of the states using the k nearest neighbors.
3. Select the skill which that state originated from.

The number of environment steps in the random walk process is $P * (T + 0.2T) * M$. Therefore, the total number of environment steps performed during the random walk process in pretraining stage is,

$$(N_{\text{initial}} + (N_{\text{initial}} + \delta) + (N_{\text{initial}} + 2 * \delta) + .. + (N_{\text{final}} - \delta)) * (T + 0.2T) * M$$

where $N_{\text{initial}}$ is the number of initial skills, $\delta$ is the number of skills to extend when most of the existing skills are converged (e.g., number of skills are $10 \rightarrow 15 \rightarrow 20 \rightarrow \ldots \rightarrow 50 \rightarrow 55$ when $N_{\text{initial}}$=10, $\delta$=5, $N_{\text{final}}$=55). Note that $N_{\text{final}}$ is the number of skills when the pretraining is ended (not predefined as a hyperparameter).

However, in long-horizon environment such as DMC (1000 timesteps), our random walk process can cause non-negligible sample inefficiencies. Thus, we further present an efficient random walk process that approximates an original random walk process with a simple approach. The detailed process is as follows:

1. During pretraining, if the skill satisfies $q_\phi(z^i|s_T) < \epsilon$ in the terminal state $s_T$ (i.e., which is not an apprentice skill), additionally perform $R$ random walk steps and store the random arrival states at the *temporary buffer (queue)*.

2. When selecting guide skill, instead of rolling out all skills, select the state in the lowest density region among the states at the *temporary buffer*.

3. Select the skill which that state originated from.

Since there are no additional environment steps when selecting the guide skill, the efficient random walk process can significantly reduce the number of environment steps compared to the original version (added environment steps are $0.2T$ per trajectory (only if its skill satisfies accuracy threshold), whereas original random walk process requires $P \times (T + 0.2T) \times M$). In Fig 7 and Table 4, we empirically show that the efficient random walk process still achieves comparably good scores on DMC.

# F    FULL RESULTS ON DEEPMIND CONTROL SUITE TASKS

Table 4: Performance comparison of DISCO-DANCE and baselines on DMC. Bold scores indicate the best model performance and underlined scores indicate the second best. The results are averaged over 9 random seeds with a standard deviation.

| Models | Cheetah | | | | Quadruped | | | | Avg |
|---|---|---|---|---|---|---|---|---|---|
| | Run | Run Backward | Flip | Flip Backward | Jump | Run | Stand | Walk | |
| Random Init | 399.80±67.55 | 384.82±17.46 | **700.02±17.87** | **715.07±9.26** | 383.41±210.83 | 150.40±80.84 | 450.59±180.30 | 147.74±209.65 | 416.48 |
| DIAYN | 394.57±104.86 | 412.92±10.96 | 642.96±60.67 | 534.23±112.17 | 278.22±133.28 | 261.78±139.92 | 512.27±209.39 | 344.41±256.60 | 422.67 |
| SMM | 389.47±125.42 | **455.61±26.24** | 633.63±66.35 | 513.66±68.78 | 215.52±254.95 | 99.81±98.38 | 289.02±257.68 | 106.83±99.95 | 337.94 |
| APS | **612.72±32.95** | 427.14±117.07 | 622.92±60.41 | 683.66±60.86 | 407.71±341.72 | 174.82±177.37 | 658.20±362.10 | 386.52±353.54 | 496.79 |
| DISDAIN | 22.15±9.89 | 18.74±6.73 | 283.08±25.22 | 269.58±31.58 | **626.26±99.13** | 285.43±121.70 | **820.57±151.74** | **479.03±162.28** | 350.60 |
| DISCO-DANCE | 521.79±85.73 | 418.25±16.07 | 671.44±50.56 | 599.48±50.81 | 612.88±193.82 | **331.32±147.09** | 700.89±283.58 | 339.17±324.73 | **524.40** |

Table 4 summarizes the full results for each task in DMC. We can observe that DISCO-DANCE gets first or second performance in seven out of eight tasks. In contrast, other baselines show leading performance only for a few tasks (i.e., DISDAIN underperforms in cheetah domain tasks). We can find out DISCO-DANCE outperforms other baselines and *random init* in terms of average performance, and this indicates that DISCO-DANCE learned more general behaviors through guidance.

# G    COMPARISON WITH UPSIDE

DISCO-DANCE and UPSIDE (Kamienny et al., 2021) have similar motivation, alleviating the inherent pessimism problem. Both of their strategies can be summarized as **"agent guides itself"** (i.e., existing skills help new/unconverged skills to explore). In DISCO-DANCE, guide skill encourages apprentice skills (unconverged skills) to explore unexplored states by minimizing the KL divergence between guide skill and apprentice skills. While DISCO-DANCE learns a set of single skills, UPSIDE learns tree-structured policy which is composed of multiple skill segments.

Fig 12 illustrates an example where a total of 8 skills are learned. When learning the policy in the unsupervised pretraining stage, UPSIDE (1) selects the skill with the largest discriminator accuracy among the leaf node skills (e.g., $z = 4$), (2) adds new skills as its children nodes (e.g., $z = 7, 8$), (3) freeze the parent skills (e.g., $z = 1, 4$ are not trained) and only train newly added skills, and (4) iteratively repeat these procedures to construct the tree-structured policies (Fig 12(b)). This makes the fundamental difference between DISCO-DANCE and UPSIDE, that **UPSIDE requires sequential execution from ancestors' skills to child skill in a top-down manner** due to its tree-structured skill policies. For example, if we want to run skill $z = 8$, UPSIDE needs to execute skills sequentially ($z = 1 \rightarrow 4 \rightarrow 8$) for $T$ timesteps, where $T$ is a hyperparameter that decides how dense the generated tree will be.

One of the key objectives of unsupervised skill discovery is to utilize the learned skills as a useful primitive at the fine-tuning stage. **However, these sequential executions of UPSIDE bring significant inefficiency at the finetuning stage**, due to the following reason. Suppose the given downstream task is to reach the goal state as fast as possible where the distance from the goal state and total execution time is given as a penalty (Fig 12). Since $z = 8$ is the skill with minimum distance from the goal, DISCO-DANCE selects skill $z = 8$ and finetunes $\pi(a|s, z = 8)$ to optimize given reward functions (Fig 12(a)). However, for UPSIDE to finetune $z = 8$, we need to finetune $z = 1, 4, 8$

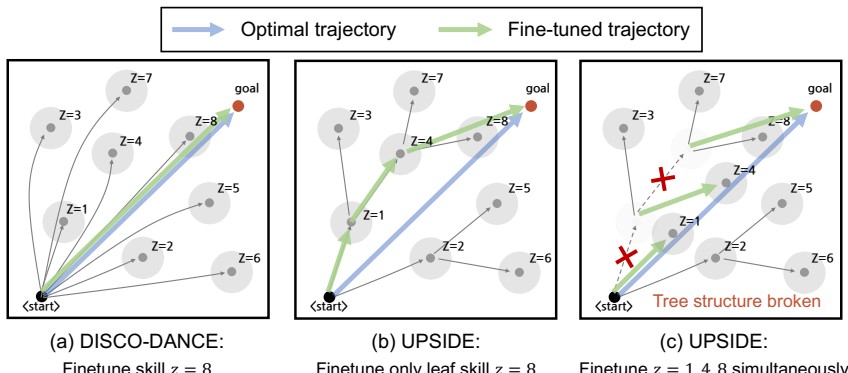

Figure 12: **Conceptual illustration of (1) learned skills of DISCO-DANCE and UPSIDE, and (2) goal-reaching downstream task.** Black edge represents skill and black circle represents the terminal states of its skill. (a) After unsupervised pretraining, DISCO-DANCE obtains a set of skills that can be used individually without the need for each skill to be used together. In finetuning stage, DISCO-DANCE selects skill $z = 8$ (i.e., skill with highest downstream task reward) and finetunes the skill to reach goal fastly. (b) However, to execute $z = 8$, UPSIDE requires sequential execution of its all ancestors skills (i.e., executing $z = 1, 4, 8$ sequentially). In finetuning stage, UPSIDE is not able to learn optimal policy since the ancestors skills are kept fixed during finetuning. (c) If we finetune ancestors skills simultaneously, the tree structure learned in pretraining phase will be broken, which makes finetuning unstable and difficult.

simultaneously (Fig 12(c)). In that case, the tree-structured skill policies learned in the pretraining stage is broken during the finetuning stage. That is, the dictionary $\{`z = 8' : [1, 4, 8]\}$ cannot be used anymore because executing $z = 1$ for $T$ timesteps does not move the agent to the original $z = 1$ terminal nodes (red $\times$ in Fig 12(c)). This means that the skill tree learned in pretraining can no longer be used, leading to ineffecient finetuning.

To avoid this limitation, UPSIDE only finetuned the leaf skill in the original paper (i.e., freeze ancestors $z = 1, 4$ and only finetunes $z = 8$ in Fig 12(b)). However, this would be still ineffective because the fixed ancestors $z = 1, 4$ are not the optimal solution to solve the given downstream task (green lines from <start> to $z = 4$ nodes in Fig 12(b)). The problem becomes more serious in a long-horizon environment such as DMC (where horizon is 1000) if we freeze all the pretrained ancestors skills and finetune only the last leaf skill. In contrast, DISCO-DANCE selects a single skill to finetune which can be executed independently, so it does not suffer from the above problem.

## H    LIMITATIONS AND FUTURE DIRECTIONS

DISCO-DANCE may cause cost (sample) inefficiency when measuring the density of the state distribution in environment with high dimensional input, such as control from pixel observations. In this case, we could utilize the downscaling technique (e.g., *cell representation* in Go-Explore) to reduce computational costs.

For the guide skill selection process to work stably, the termination state should be reliably reached. Therefore, when the stochasticity of the environment is too large, it would not be easy to select a guide skill only with the simple random walk process we proposed. We leave it as future work for alleviating such difficulties.

# I   VISUALIZATION OF LEARNED SKILLS

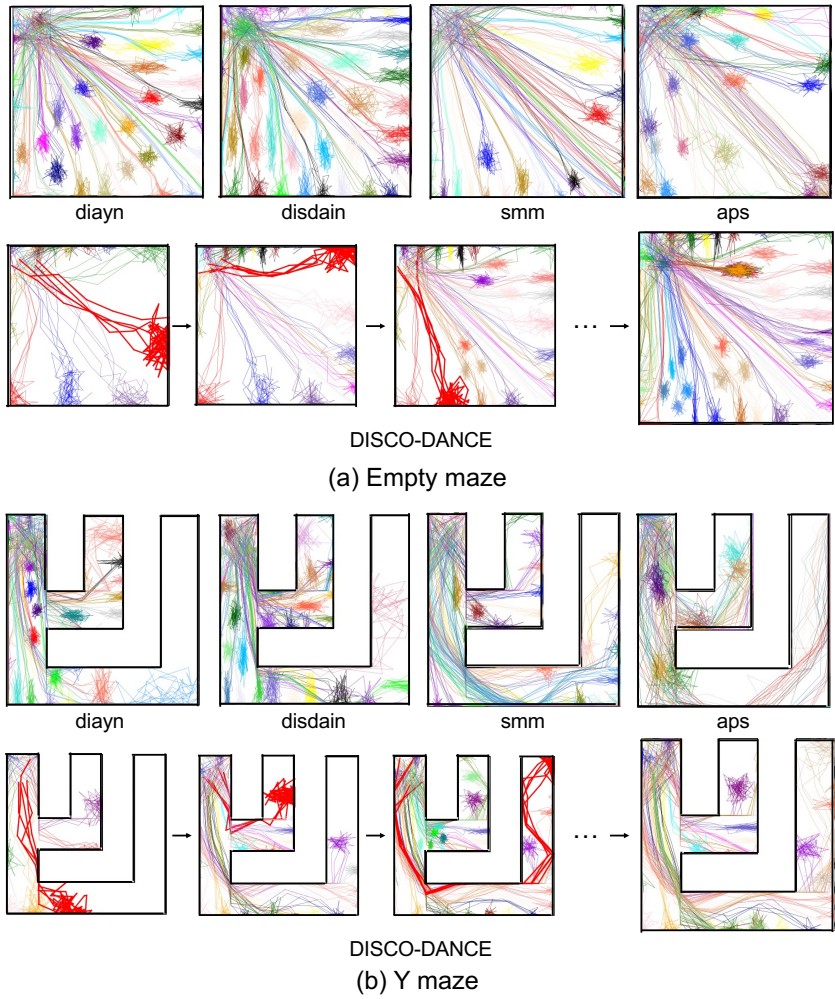

Figure 13: **Qualitative visualization of the learned skills on Emtpy maze and Y maze.** Visualization of multiple rollouts of learned skills by baseline models. For DISCO-DANCE, we visualize our curriculum precedure during training. Bold red lines indicate the guide skill.

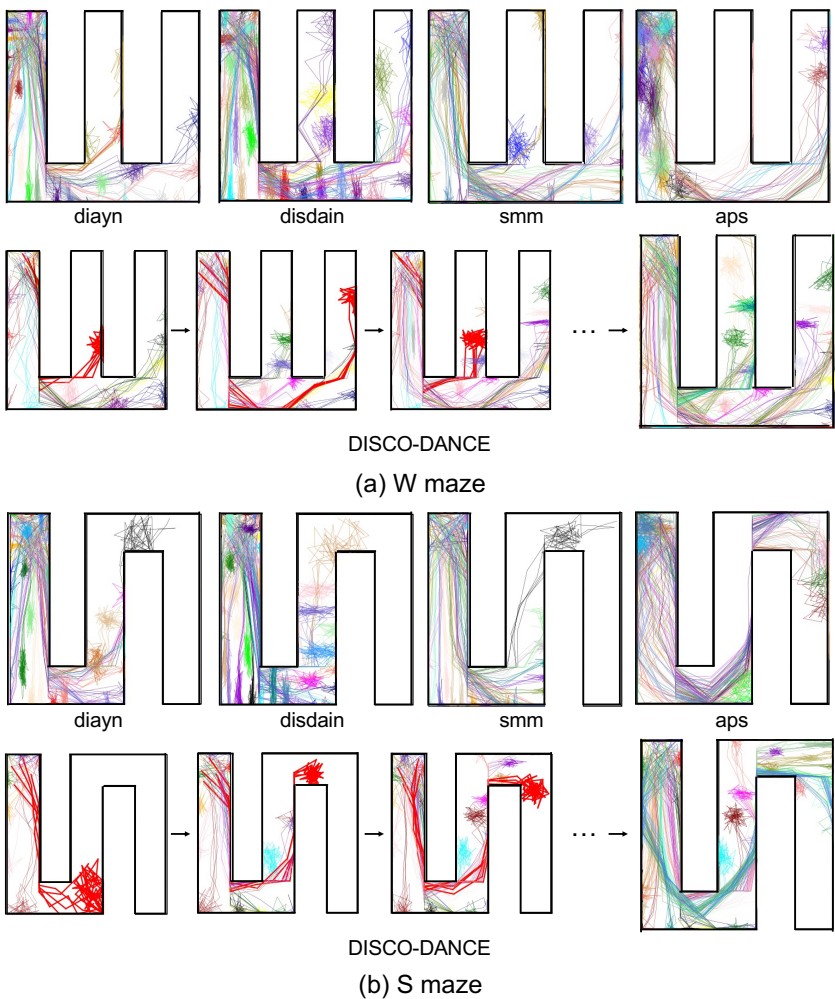

Figure 14: **Qualitative visualization of the learned skills on W maze and S maze.** Visualization of multiple rollouts of learned skills by baseline models. For DISCO-DANCE, we visualize our curriculum precedure during training. Bold red lines indicate the guide skill.

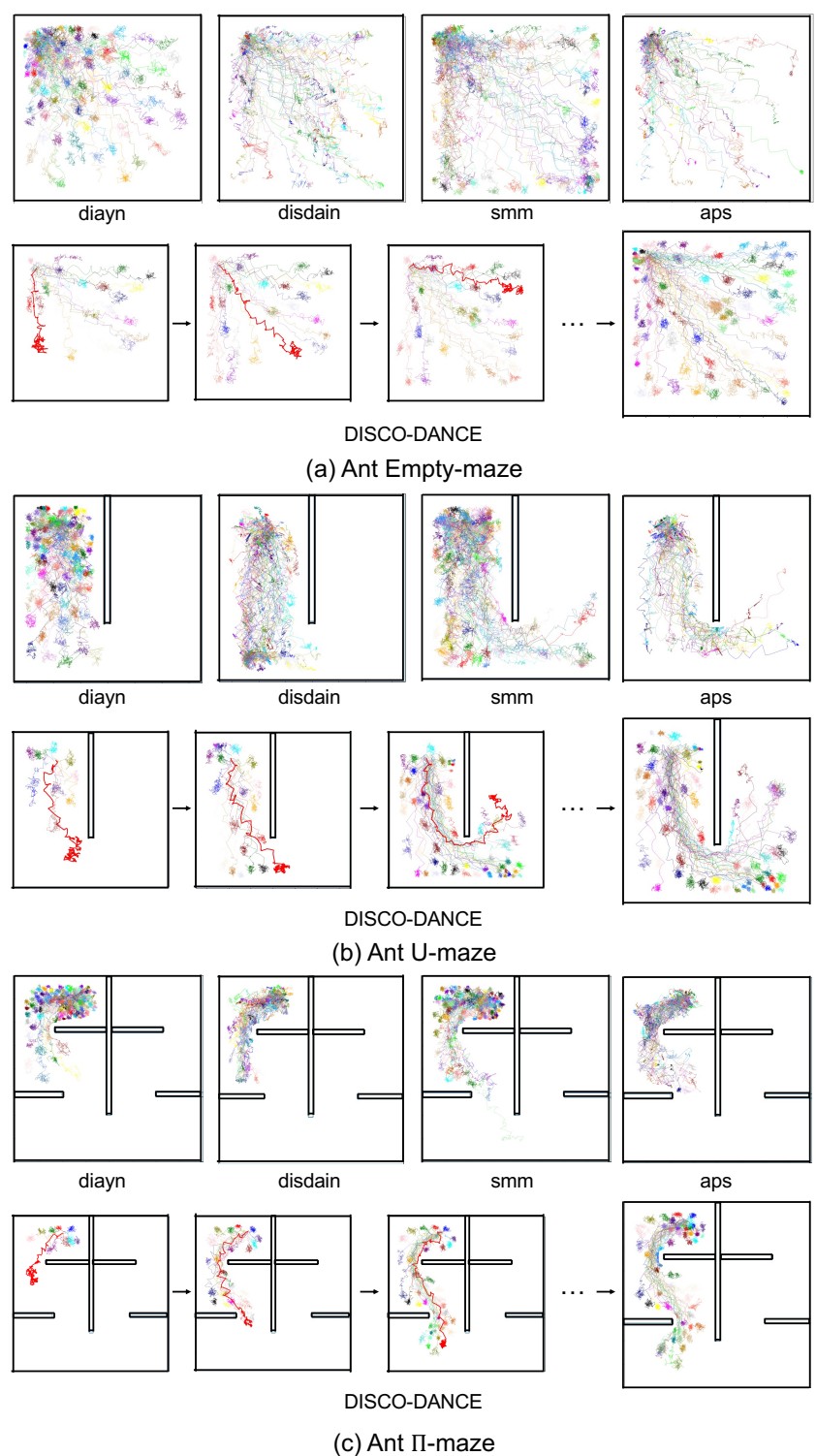

Figure 15: **Qualitative visualization of the learned skills on Ant Empty-maze, Ant U-maze, and Ant Π-maze.** Visualization of multiple rollouts of learned skills by baseline models. For DISCO-DANCE, we visualize our curriculum precedure during training. Bold red lines indicate the guide skill.

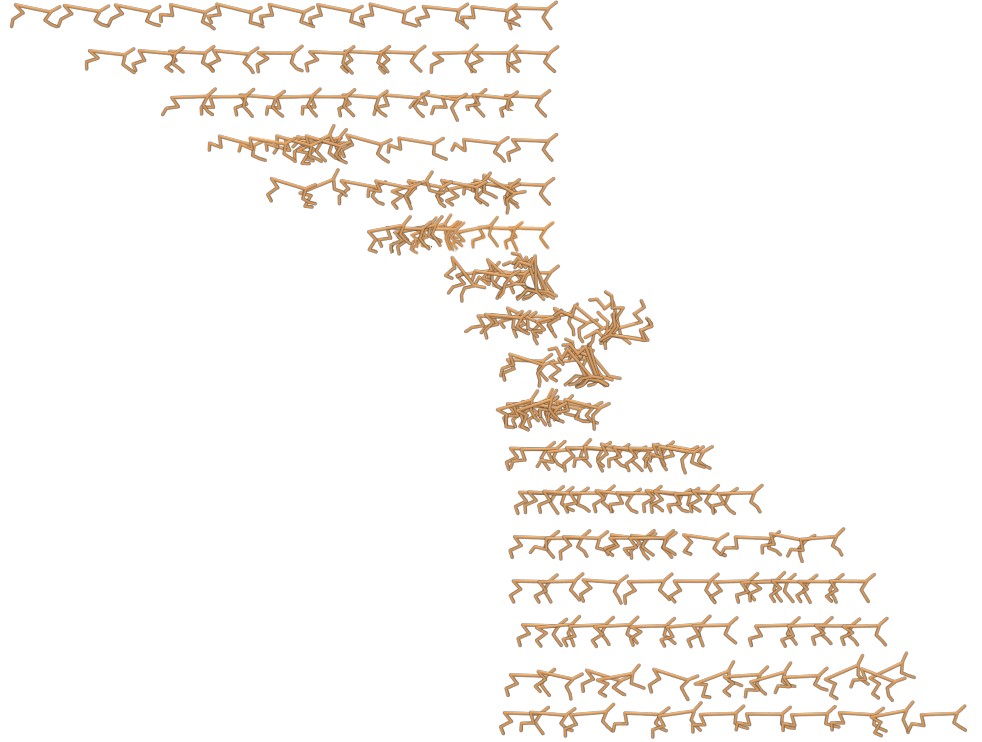

Figure 16: **Qualitative visualization of the learned skills in cheetah domain in DMC.** DISCO-DANCE learned 100 skills without reward function. To effectively show how diverse skills DISCO-DANCE learned, we select 17 skills based on x coordinate of each skill's terminal state.

