# OpenReview forum: "DISCO-DANCE: Learning to Discover Skills with Guidance"
_ICLR.cc/2023/Conference — Submitted to ICLR 2023_

### Official Review · Reviewer_RjRU · 2022-10-20

**Confidence:** 4
**Correctness:** 3
**Technical Novelty And Significance:** 2
**Empirical Novelty And Significance:** 2
**Recommendation:** 3

**Clarity, Quality, Novelty And Reproducibility:**

Clarity: The paper is generally easy to follow and clearly written, but I have a few questions still:
- A minor clarification concerns the purpose of underscoring in results tables
- Finally, it would be good to provide videos of the skill policies in the continuous-control tasks considered

Quality: Besides further baselines as mentioned above, and additional discussion of limitations of specific design choices and hyper-parameters, there are few minor points wrt experiments:
- The time horizon for the Ant maze environments is pretty short (200 steps). Would your method produce skills that cover the whole maze (Figure 14 b and c) with a sufficiently long time horizon?
- Likewise on Ant Maze, for downstream learning you provide a dense reward that takes walls into account. What's the performance of the random-init policy here?

Novelty: In terms of overall motivation, there seems to be a strong relation to UPSIDE (identifying skills that provide good final states for further exploring the environment) which is not discussed in detail. Why would your approach be preferred?

Reproducibility: The authors use publicly available benchmarks, and it seems like Algorithm 1 can be tucked on top of existing DIAYN implementations. A few sentences on `find_guide_skill` would be appreciated; in 3.1., you mention random walks from skill termination states but there are a few possible ways this could be done concretely. In 4.2.2, the authors note that they follow the fine-tuning procedure in Liu & Abbeel (2021), which could be explained in a few words in the Appendix (at least, I couldn't find the details I was looking for with a cursory glance at that reference).

**Strength And Weaknesses:**

Strengths:
- The paper is well-written with a clear presentation and motivation
- The exploration problem is quite pressing in unsupervised skill discovery (USD), and failure modes of previous works are nicely summarized in Figure 1.
- Although the approach is primarily motivated in navigation environments, it also achieves comparably good scores on DMC-100k.

Weaknesses:
- UPSIDE would be a very relevant baseline here since its main motivation (improving exploration by extending skills) is very related. I'm not 100% sure whether the environments are matching, but what about adding the results from Table 2 in Kamienny et al. (2021) to your Table 1? UPSIDE achieves a coverage of 85.67 in the bottleneck maze, which (assuming the environments are the same), almost matches your result.
- Another potentially USD baseline would be the training of a goal-based policy, e.g., with hindsight experience replay. This should be particularly effective in navigation environments, assuming the extents of the maze are known.
- The goal skill selection seems to be mainly based on heuristics, assuming that (1) skill termination states can be reliably reached and (2) random walks. It would good to (briefly) discuss possible failure modes here, e.g., in large state spaces (I assume that in the Ant mazes, you consider X/Y only for the random walk end states?), for stochastic environments, or if dynamics differ more significantly across the state region as in some Atari games.
- In a similar vein, the proposed algorithm introduces several hyper-parameters. Apart from an ablation of major components (which is useful), the sensitivity to, e.g., the accuracy threshold for which skills should be guided is not discussed.

**Summary Of The Paper:**

The paper at hand proposes a new inductive bias (in the form of a proxy reward) for MI-based unsupervised skill discovery. The reward is designed so that exploration is improved, in particular in environments with bottleneck states or non-trivial dynamics. This is turn motivated by the observation that MI-based methods generally exhibit poor exploration capabilities as it is not strictly required to satisfy the main training criterion (the ability to discriminate skills by they states they visit).

**Summary Of The Review:**

Overall, my main concern with the paper is the supposedly minor amount of novelty with respect to UPSIDE, which it is not considered as a baseline and the pros/cons of both approaches are not discussed. I would be happy to see the authors addressing this concern.

---

> ### Author Response · Authors · 2022-11-15
> **Response to reviewer RjRU (R3)**
>
> We appreciate the thorough review and thoughtful comments.
>
> **Q10. UPSIDE would be a very relevant baseline here since its main motivation (improving exploration by extending skills) is very related. I'm not 100% sure whether the environments are matching, but what about adding the results from Table 2 in Kamienny et al. (2021) to your Table 1? UPSIDE achieves a coverage of 85.67 in the bottleneck maze, which (assuming the environments are the same), almost matches your result.**
>
> A. Please refer to the general response #1.
>
> **Q11. Another potentially USD baseline would be the training of a goal-based policy, e.g., with hindsight experience replay. This should be particularly effective in navigation environments, assuming the extents of the maze are known.**
>
> A. We agree that the paper would benefit from employing goal-based policy with HER (i.e., π(a|s,z) where z is a goal state) as a baseline in navigation environments. However, since (1) this paper focuses on skill-based policies and (2) goal-based policy is not directly applicable to continuous control tasks, we only utilized skill-based policies as a baseline.
>
> **Q12. The goal skill selection seems to be mainly based on heuristics, assuming that (1) skill termination states can be reliably reached and (2) random walks. It would good to (briefly) discuss possible failure modes here, e.g., in large state spaces (I assume that in the Ant mazes, you consider X/Y only for the random walk end states?), for stochastic environments, or if dynamics differ more significantly across the state region as in some Atari games.**
>
> A. As the reviewer pointed out, we only consider x-y for random walk states, since the goal of navigation environments is to learn a set of skills that can cover most of the maze with respect to the x-y position. However, in continuous control environments (DMC-cheetah, quadruped), we consider full observable dimensions (17 for cheetah, 78 for quadruped), which is much larger than 2 (x-y position). We empirically found that our random walk process performs well in DMC. For a pixel based input (e.g., Atari games), measuring the density of the state distribution may cause cost (sample) inefficiency. In this case, we could utilize the downscaling technique (e.g., cell representation in Go-Explore [1]) to reduce computational costs.
>
> For the guide skill selection process to work stably, the termination state should be reliably reached. Therefore, when the stochasticity of the environment is too large, it would not be easy to select a guide skill only with the simple random walk process we proposed. As the reviewer mentioned, these are clear limitations of these works and leave it as future work for alleviating such difficulties. We added this discussion of these failure modes in the Appendix H.
>
> [1] Adrien Ecoffet, Joost Huizinga, Joel Lehman, Kenneth O Stanley, and Jeff Clune. First return, then explore.Nature, 2021.
>
> **Q13. In a similar vein, the proposed algorithm introduces several hyper-parameters. Apart from an ablation of major components (which is useful), the sensitivity to, e.g., the accuracy threshold for which skills should be guided is not discussed.**
>
> A. We conduct additional ablation experiments with respect to the accuracy threshold (threshold=0.3, 0.5, 0.7, 5 seeds respectively) in bottleneck maze. Please refer to [Figure 10(b)](https://drive.google.com/file/d/1D_4LlnXyRxVUTYEe4k5QqEbET9-xqcMl/view?usp=share_link) in Appendix C in the revised paper. Figure 10(b) shows that DISCO-DANCE is robust to the accuracy threshold.
>
> **Q14. A minor clarification concerns the purpose of underscoring in results tables**
>
> A. Sorry, but we did not understand the question. Could you elaborate on the question? In our experiment tables, bold scores indicate the best model performance and underlined scores indicate the second best. We added the explanation in Section 4.1.
>
> **Q15. Finally, it would be good to provide videos of the skill policies in the continuous-control tasks considered.**
>
> A. We added qualitative results (visualization of the learned skills) for DMC cheetah environment, please refer to [Figure 16](https://drive.google.com/file/d/1GjMGLFCBunSRVyBCxPScDYJdrQsFV7Au/view?usp=share_link) in Appendix I.

---

> > ### Author Response · Authors · 2022-11-15
> > **Response to reviewer RjRU (R3)**
> >
> > **Q16. The time horizon for the Ant maze environments is pretty short (200 steps). Would your method produce skills that cover the whole maze (Figure 14 b and c) with a sufficiently long time horizon?**
> >
> > A. First of all, we experimented Ant maze environments with 400 time horizons. We made a typo when writing Appendix B, and it has been corrected in an updated manuscript.
> >
> > For π-maze, which is the most difficult, the agent can reach the farthest region from the initial state within 400 steps. We can know this from the DISCO-DANCE arriving at the goal point within 400 steps in the finetuning experiments (Figure 6). This indicates that it is possible for agent to learn skills that cover the whole maze, even in π-maze. In addition, we experimented DISCO-DANCE with larger maximum timestep (>400), but there was no significant difference.
> >
> > **Q17. Likewise on Ant Maze, for downstream learning you provide a dense reward that takes walls into account. What's the performance of the random-init policy here?**
> >
> > A. We use random-init policy (in Figure 6 and Figure 7) as a baseline, to show the performance of learning from scratch (i.e., not using unsupervised pretrained weights).
> >
> > **Q18. A few sentences on find_guide_skill would be appreciated.  In 3.1, you mention random walks from skill termination states but there are a few possible ways this could be done concretely.**
> >
> > A. Please refer to General response #2.
> >
> > **Q19. In 4.2.2, the authors note that they follow the fine-tuning procedure in Liu & Abbeel (2021), which could be explained in a few words in the Appendix (at least, I couldn't find the details I was looking for with a cursory glance at that reference).**
> >
> > A. For DIAYN, SMM, DISDAIN and DISCO-DANCE (discrete skills), we finetune the skill with maximum downstream task reward (same as Ant mazes). For APS (continuous skill), we first randomly select an arbitrary skill z and rollout episodes, then solve the linear regression problem to select the task-specific skill z (following same protocol in APS paper). After the skill is selected for finetuning, for all algorithms, selected skill is fine-tuned for 100K steps. Note that this is different from URLB in that the skill selection process is not included in 100K, but this is still a fair comparison as all algorithms use the same number of interactions (samples) to select skills, except for APS (as we mentioned in “Response to all reviewers: Correction in experiemental results”, APS ran fewer environment steps compared to other algorithms). Therefore we re-finetuned APS using the same number of environment steps as other algorithms, and reported the performance in Figure 7 and Table 4. We added the detailed explanation of finetuning in DMC and clarified that skill selection process is not involved in 100K, in Section 4.3 and Appendix B.1.

---

> ### Author Response · Authors · 2022-12-12
> **A gentle reminder to Reviewer RjRU (R3)**
>
> Dear reviewer RjRU (R3),
>
> We are confident that your insightful comments have made our paper more solid.
>
> Kindly let us know if you have any further comments or suggestions.
>
> Best,
>
> Paper 2527 authors

---

### Official Review · Reviewer_DqoP · 2022-10-21

**Confidence:** 4
**Correctness:** 4
**Technical Novelty And Significance:** 3
**Empirical Novelty And Significance:** Not applicable
**Recommendation:** 5

**Clarity, Quality, Novelty And Reproducibility:**

### Clarity
The paper is easy to understand and clearly written.

### Quality
- Appendix B.1: taks -> tasks

### Novelty
The proposed method is novel to my knowledge.

### Reproducibility
The authors have not released the code yet (but claimed to release it when the discussion period begins).

**Strength And Weaknesses:**

### Strengths
- The authors tackle an important problem of skill discovery with a sensible approach.
- The paper is clearly written and the figures are helpful for understanding the concept of the method.

### Weaknesses
- There is a missing baseline, UPSIDE. Since UPSIDE uses a very similar strategy (they also have a "directed" part and "random-walk" part as in this work) and shows significantly improved coverage in the same 2D maze domain, I believe the authors should include comparisons with UPSIDE and report the results.
- The proposed method seems to leverage a fairly strong assumption that the environment is resettable to any arbitrary state (for performing multiple random walks from the terminal state of each skill). Is there any efficient way to avoid using this assumption, and how well does the method perform without it? For example, just running each skill $R$ times and then doing a single random walk after each of the rollouts could be one naive way to resolve this, but this strategy may cause significant inefficiency in terms of the number of environment steps. Also, how do the authors take this (the number of samples required for random walks) into account when computing the number of total training steps (e.g., 5M steps for hard mazes)?
- It seems that the method is tailored toward synthetic 2D maze environments. What kind of skills does DISCO-DANCE learn in the DMC HalfCheetah and Quadruped environments? Videos and figures of learned skills in such environments would be very helpful to further understand the method.
- The result in Table 2 is only computed from three random seeds. Given its marginal performance improvement and the overlapping confidence intervals, it would be difficult to say DISCO-DANCE shows superior performance to the baselines in this environment.

### Additional questions
- Why is the performance of DISDAIN in Table 4 significantly worse than DIAYN in HalfCheetah? According to the DISDAIN paper, DISDAIN with $N = 1$ and $\lambda = 0$ (in their notations) exactly recovers the standard DIAYN, so I believe its performance should be at least equal to DIAYN's.
- How do the authors decide to add new skills versus reuse existing ones?
- How do the authors fine-tune their agents (and other baselines) on the DMC benchmark?


**Summary Of The Paper:**

The paper suggests an unsupervised skill discovery method named DISCO-DANCE. The authors focus on the issue that the commonly used MI reward ($I(S; Z)$) does not encourage exploration, and propose a novel approach to resolve this problem. Their method (DISCO-DANCE) first finds a "guide" skill that likely leads to unexplored states and then learns several "apprentice" skills that explore near the terminal state of the guide skill. They evaluate DISCO-DANCE on 2D maze and continuous control environments, showing that their proposed method outperforms previous skill discovery methods, such as DIAYN, SMM, APS, and DISDAIN.

**Summary Of The Review:**

While the method tackles a relevant problem of skill discovery and shows improved performance, due to insufficient evaluations and comparisons, I am not able to recommend acceptance.

(11/18 Update): While I appreciate the response as well as the effort to make comparisons with UPSIDE, I believe a fair comparison between DISCO-DANCE and its closest baseline (UPSIDE) is necessary for assessing this work. I acknowledge that it could be difficult to reproduce the baseline's result without its code, but I feel it is a bit premature to conclude that UPSIDE is not reproducible within this short period of the discussion phase. Therefore, I cannot recommend acceptance in its current form.

---

> ### Author Response · Authors · 2022-11-15
> **Response to reviewer DqoP (R2)**
>
> We greatly appreciate your valuable comments and feedback.
>
> **Q3. There is a missing baseline, UPSIDE. Since UPSIDE uses a very similar strategy (they also have a "directed" part and "random-walk" part as in this work) and shows significantly improved coverage in the same 2D maze domain, I believe the authors should include comparisons with UPSIDE and report the results.**
>
> A. Please refer to general response #1.
>
> **Q4. The proposed method seems to leverage a fairly strong assumption that the environment is resettable to any arbitrary state (for performing multiple random walks from the terminal state of each skill). Is there any efficient way to avoid using this assumption, and how well does the method perform without it? For example, just running each skill  R times and then doing a single random walk after each of the rollouts could be one naive way to resolve this, but this strategy may cause significant inefficiency in terms of the number of environment steps. Also, how do the authors take this (the number of samples required for random walks) into account when computing the number of total training steps (e.g., 5M steps for hard mazes)?**
>
> A. Please refer to general response #2.
>
> **Q5. It seems that the method is tailored toward synthetic 2D maze environments. What kind of skills does DISCO-DANCE learn in the DMC HalfCheetah and Quadruped environments? Videos and figures of learned skills in such environments would be very helpful to further understand the method.**
>
> A. We added qualitative results (visualization of the learned skills) for DMC cheetah environment, please refer to [Figure 16 in Appendix I](https://drive.google.com/file/d/1GjMGLFCBunSRVyBCxPScDYJdrQsFV7Au/view?usp=share_link).
>
> **Q6. The result in Table 2 is only computed from three random seeds. Given its marginal performance improvement and the overlapping confidence intervals, it would be difficult to say DISCO-DANCE shows superior performance to the baselines in this environment.**
>
> A.  We conducted additional experiments with 2 more seeds. We updated the Table 2 with 5 seeds. Table 2 demonstrates that DISCO-DANCE is superior to other baselines in terms of state coverage.
>
> **Q7. Why is the performance of DISDAIN in Table 4 significantly worse than DIAYN in HalfCheetah? According to the DISDAIN paper, DISDAIN with N=1 and lambda=0 (in their notations) exactly recovers the standard DIAYN, so I believe its performance should be at least equal to DIAYN's.**
>
> A. As reviewer pointed out, DISDAIN with N=1 and lambda=0 exactly recovers DIAYN. However, to take advantage of DISDAIN’s characteristic of giving uncertainty between discriminators as a bonus reward, we grid-searched the number of ensembles N from {5, 10, 20, 40} (not searched N=1, lambda=0). In DISDAIN paper, they utilize N as 2 for Four-Rooms (toy problem) and 40 for Atari games. Since the complexity of DMC’s input observation space is in between Four-Rooms and Atari, we believe that searching between 5 and 40 is reasonable.
>
> We speculate that lower performance of DISDAIN in Cheetah comes from its lower sample efficiency in the pretraining stage. In DISDAIN paper, it takes about 270M steps for DISDAIN to learn more skills than DIAYN even in a toy problem (Four-Rooms). Table 2 also shows DISDAIN’s lower state-coverage than all other baselines (training timestep is 5M). We conjecture that simultaneously training N discriminators and skill policies cause inefficiency in training.

---

> > ### Author Response · Authors · 2022-11-15
> > **Response to reviewer DqoP (R2)**
> >
> > **Q8. How do the authors decide to add new skills versus reuse existing ones?**
> >
> > A. As in Algorithm 1 in Appendix A, if most existing skills are discriminable enough, we add new skills. Specifically, if the number of skills whose $q_\phi(z^i|s) < \epsilon$ is less than $\rho$ ($\epsilon$ is the same as in line 9 of Algorithm 1), we add  $\rho$ skills. We set $\rho$ for hyper-parameter. We corrected the typo in Appendix A and B.
> >
> > **Q9. How do the authors fine-tune their agents (and other baselines) on the DMC benchmark?**
> >
> > A. For DIAYN, SMM, DISDAIN and DISCO-DANCE (discrete skills), we finetune the skill with maximum downstream task reward (same as Ant mazes). For APS (continuous skill), we first randomly select an arbitrary skill z and rollout episodes, then solve the linear regression problem to select the task-specific skill z (following same protocol in APS paper). After the skill is selected for finetuning, for all algorithms, selected skill is fine-tuned for 100K steps. Note that this is different from URLB in that the skill selection process is not included in 100K, but this is still a fair comparison as all algorithms use the same number of interactions (samples) to select skills, except for APS (as we mentioned in “Response to all reviewers: Correction in experiemental results”, APS ran fewer environment steps compared to other algorithms). Therefore we re-finetuned APS using the same number of environment steps as other algorithms, and reported the performance in Figure 7 and Table 4. We added the detailed explanation of finetuning in DMC and clarified that skill selection process is not involved in 100K, in Section 4.3 and Appendix B.1.

---

### Official Review · Reviewer_xuUS · 2022-10-25

**Confidence:** 3
**Correctness:** 3
**Technical Novelty And Significance:** 2
**Empirical Novelty And Significance:** 2
**Recommendation:** 5

**Clarity, Quality, Novelty And Reproducibility:**

The paper is clear, the visualizations are informative and helpful.  The novelty of the Disco-dance is hard to assess given similarities to prior work mentioned above and little direct comparison with those approaches.

**Strength And Weaknesses:**

The paper addresses an important problem in skill learning, which is the diversity of states reached by current unsupervised rewards.  The Disco-Dance method is very well described, and experiments are clear as well.

My major concerns:

- Similarity to previous methods is difficult to assess.  Disco-Dance seems very similar to both Go-Explore and Direct-then-Diffuse (UPSIDE) both of which are cited in the current paper.  However, despite the citations, there is little discussion of the differences between Disco-Dance and these prior approaches, and disappointingly there is no experimental comparison to those previous approaches.  This makes the impact of the current paper hard to assess.

- In high-dimensional state spaces, like in the ant-maze experiments, how is the distance measured?  Does the approach require knowledge of which subspace represents the x-y plane?  If not, I'd imagine the number of skills required to get the coverage numbers reported in table 2 would be very large.  If so, does that not defeat the purpose of having high-dimensional state spaces, at least for the concerns of the current approach?

**Summary Of The Paper:**

Disco-Dance is a method for improving skill learning by expanding the diversity of the states that skills can reach.  The method first selects guide skills by finding skills most likely to lead to new states, then trains new skills (or old skills that have low discriminability) to reach novel states using mutual-information maximization.

**Summary Of The Review:**

An interesting method for dealing with a difficult problem.  My concerns mostly involve similarity to previous work without comparison to those approaches.  If the differences can be clarified, both through explanation and experiment, I would be willing to raise my score.

---

> ### Author Response · Authors · 2022-11-15
> **Response to reviewer xuUS (R1)**
>
> Thank you for your time and thoughtful comments.
>
> **Q1. Similarity to previous methods is difficult to assess. Disco-Dance seems very similar to both Go-Explore and Direct-then-Diffuse (UPSIDE) both of which are cited in the current paper. However, despite the citations, there is little discussion of the differences between Disco-Dance and these prior approaches, and disappointingly there is no experimental comparison to those previous approaches. This makes the impact of the current paper hard to assess.**
>
> A. Please refer to general response #1.
>
> **Q2. In high-dimensional state spaces, like in the ant-maze experiments, how is the distance measured? Does the approach require knowledge of which subspace represents the x-y plane? If not, I'd imagine the number of skills required to get the coverage numbers reported in table 2 would be very large. If so, does that not defeat the purpose of having high-dimensional state spaces, at least for the concerns of the current approach?**
>
> A. Since the goal of **navigation environments** in unsupervised pretraining stage is to learn a set of skills that can cover most of the maze with respect to x-y plane, we restrain the input of the discriminator as x,y coordinates and measure the state coverage with how much the skills fill the maze’s x-y plane (Table 1,2). Specifically, the x,y axes of each maze is discretized into 10 buckets (total 100 buckets), and state coverage is measured by the number of bins reached by learned skills.
>
> On the other hand, in DMC, its objective is to learn a set of skills that can be utilized as useful primitives for various downstream **control tasks** (e.g., jump, run backward, stand). Therefore, all observable dimensions are passed to the discriminator (18 dimension for cheetah and 78 dimension for quadruped). Thus, in DMC, the agent can learn a various set of skills that reach different state space regions with respect to every observable dimensions (not just x-y plane).
>
> As the reviewer pointed out, number of skills required to achieve high state coverage would be very large (there are total 10^78 bins in quadruped). Therefore, the downstream task performance is utilized as a main metric in DMC.

---

> ### Author Response · Authors · 2022-12-12
> **Kind reminder for discussion**
>
> Dear reviewer xuUS (R1),
>
> We deeply appreciate your valuable feedback on improving our paper.
>
> We are looking forward to receiving any further comments or suggestions.
>
> Best,
>
> Paper 2527 authors

---

### Author Response · Authors · 2022-11-15
**General response #2: Random walk process.**

**(R2) Q4. The proposed method seems to leverage a fairly strong assumption that the environment is resettable to any arbitrary state (for performing multiple random walks from the terminal state of each skill). Is there any efficient way to avoid using this assumption, and how well does the method perform without it? For example, just running each skill R times and then doing a single random walk after each of the rollouts could be one naive way to resolve this, but this strategy may cause significant inefficiency in terms of the number of environment steps. Also, how do the authors take this (the number of samples required for random walks) into account when computing the number of total training steps (e.g., 5M steps for hard mazes)?**

**(R3) Q18. A few sentences on find_guide_skill would be appreciated.  In 3.1, you mention random walks from skill termination states but there are a few possible ways this could be done concretely.**

A. The detailed description of random walk process is as follows:

1. After performing rollout for all P learned skills till the terminal state, we perform R random walks and collect P*R number of random arrival states (R is a hyper-parameter and we set R as 0.2 * time horizon T for all environments). Repeat this process a total of M times.
2. Pinpoint the state in the lowest density region among the P*R*M number of states. We measure the density of the states using the k nearest neighbors.
3. Select the skill which that state originated from.

The number of environment steps in the random walk process is P*(T+0.2T)*M. Therefore, the total number of environment steps performed during the random walk process in pretraining stage is,

$(N_\text{initial} + (N_\text{initial} + 𝛿) + (N_\text{initial} + 2*𝛿) + .. + (N_\text{final} - 𝛿)) *(T + 0.2T)*M$

where $N_\text{initial}$ is the number of initial skills and 𝛿 is the number of skills to extend when most of the existing skills are converged (e.g.,  number of skills are **10 -> 15 -> 20 -> … -> 50 -> 55** when $N_\text{initial}$=10, 𝛿=5, $N_\text{final}$=55). Note that $N_\text{final}$ is the number of skills when the pretraining is ended (not predefined as a hyperparameter).

For experimental evaluation, we mistakenly did not include these random walk steps in the total pretraining steps. Therefore, for 2D Mazes and Ant Mazes, **we have updated Table 1,2 with the checkpoints of each pretrained model with earlier timesteps** so that the total pretraining time is no more than 2M/5M, including random walk steps. For example, 2D bottleneck took 135,000 random walk steps during the training (=(30+35+...+70)*(50+10)*5), so we updated the state coverage with model checkpoints at t=4,500,000 (< 5M - 135,000). We clarified how we chose the checkpoint in the revised manuscript (Appendix A). Even after updating the results, it is shown that DISCO-DANCE still outperforms other baselines, especially as the environments become more complex. **We also updated the finetuning experiments (Figure 6)** (finetune with these checkpoints). Figure 6 shows DISCO-DANCE still outperforms baselines even in finetuning experiments.

As reviewer 2 pointed out, in environments with large maximum timesteps such as DMC (1000 timesteps), our random walk process can cause non-negligible sample inefficiencies because P*(T+0.2T)*M increases linearly as T increases. Thus, we further present an efficient random walk process that approximates an original random walk process with a simple approach. The detailed process is as follows:

1. During pretraining, if the skill satisfies $q_\phi(z^i|s_T) < \epsilon$ in the terminal state $s_T$ (i.e., which is not an apprentice skill),  additionally perform R random walk steps and store the random arrival states at the temporary buffer (queue).
2. When selecting the guide skill, instead of rolling out all skills, select the state in the lowest density region among the states at the temporary buffer.
3. Select the skill which that state originated from.

Since there are no additional environment steps when selecting the guide skill, the efficient random walk process can significantly reduce the number of environment steps compared to the original version (added environment steps are **0.2T** per trajectory (only if its skill satisfies accuracy threshold), whereas original random walk process requires **P*(T+0.2T)*M**). By using this efficient random walk process, **we updated the downstream task performances (Figure 7, Table 4) of DISCO-DANCE in DMC**, which is trained for 2M steps including the efficient random walk process. We empirically show that the efficient random walk process still achieves comparably good scores on DMC, in Table 4. We added a detailed description of the efficient random walk process in the Appendix E.

---

### Author Response · Authors · 2022-11-15
**General response #1: Comparison with UPSIDE (1).**

**(R1) Q1. Similarity to previous methods is difficult to assess. Disco-Dance seems very similar to both Go-Explore and Direct-then-Diffuse (UPSIDE) both of which are cited in the current paper. However, despite the citations, there is little discussion of the differences between Disco-Dance and these prior approaches, and disappointingly there is no experimental comparison to those previous approaches. This makes the impact of the current paper hard to assess.**

**(R2) Q3. There is a missing baseline, UPSIDE. Since UPSIDE uses a very similar strategy (they also have a "directed" part and "random-walk" part as in this work) and shows significantly improved coverage in the same 2D maze domain, I believe the authors should include comparisons with UPSIDE and report the results.**

**(R3) Q10. UPSIDE would be a very relevant baseline here since its main motivation (improving exploration by extending skills) is very related. I'm not 100% sure whether the environments are matching, but what about adding the results from Table 2 in Kamienny et al. (2021) to your Table 1? UPSIDE achieves a coverage of 85.67 in the bottleneck maze, which (assuming the environments are the same), almost matches your result.**

A. As all the reviewers pointed out, DISCO-DANCE and UPSIDE have similar motivation, alleviating the inherent pessimism problem. Both of their strategies can be summarized as **“agent guides itself”** (i.e., existing skills help new/unconverged skills to explore). In DISCO-DANCE, guide skill encourages apprentice skills (unconverged skills) to explore unexplored states by minimizing the KL divergence between guide skill and apprentice skills. While DISCO-DANCE learns a set of single skills, UPSIDE learns tree-structured policy which is composed of multiple skill segments. We visualize the skills learned with DISCO-DANCE and UPSIDE [in this figure](https://drive.google.com/file/d/1CS_4IyMKj50uE2fkFlXQusknljYTxX1s/view?usp=share_link) (we added this figure in Appendix G).

[Figure 12](https://drive.google.com/file/d/1CS_4IyMKj50uE2fkFlXQusknljYTxX1s/view?usp=share_link) illustrates an example where a total of 8 skills are learned. When learning the policy in the *unsupervised pretraining stage*, UPSIDE (1) selects the skill with the largest discriminator accuracy among the leaf node skills (e.g., z=4), (2) adds new skills as its children nodes (e.g., z=7,8), (3) freeze the parent skills (e.g., z=1,4 are not trained) and only train newly added skills, and (4) iteratively repeat these procedures to construct the tree-structured policies (Figure 12(b)). This makes the fundamental difference between DISCO-DANCE and UPSIDE, that **UPSIDE requires sequential execution from ancestors’ skills to child skill in a top-down manner** due to its tree-structured skill policies. For example, if we want to run skill z=8, UPSIDE needs to execute skills sequentially (z=1->4->8)  for T timesteps, where T is a hyperparameter that decides how dense the generated tree will be.

One of the key objectives of unsupervised skill discovery is to utilize the learned skills as a useful primitive at the fine-tuning stage. However, **these sequential executions of UPSIDE bring significant inefficiency at the *finetuning stage*,** due to the following reason. Suppose the given downstream task is to reach the goal state as fast as possible where the distance from the goal state and total execution time is given as a penalty (Figure 12). Since z=8 is the skill with minimum distance from the goal, DISCO-DANCE selects skill z=8 and finetunes π(a|s,z=8) to optimize given reward functions (Figure 12(a)). However, for UPSIDE to finetune z=8, we need to finetune z=1,4,8 simultaneously (Figure 12(c)). In that case, the tree-structured skill policies learned in the pretraining stage is broken during the finetuning stage. That is, the dictionary {‘z=8’: [1,4,8]} cannot be used anymore because executing z=1 for T timesteps does not move the agent to the original z=1 terminal nodes (red x in Figure 12(c)). This means that the skill tree learned in pretraining can no longer be used, leading to ineffecient fine-tuning.

To avoid this limitation, UPSIDE only finetuned the leaf skill in the original paper (i.e., freeze ancestors z=1,4 and only finetunes z=8 in Figure 12(b)). However, this would be still ineffective because the fixed ancestors z=1,4 are not the optimal solution to solve the given downstream task (green lines from <start> to z=4 nodes in Figure 12(b)). The problem becomes more serious in a long-horizon environment such as DMC (where horizon is 1000) if we freeze all the pretrained ancestors skills and finetune only the last leaf skill. In contrast, DISCO-DANCE selects a single skill to finetune which can be executed independently, so it does not suffer from the above problem.

---

> ### Author Response · Authors · 2022-11-15
> **General response #1: Comparison with UPSIDE (2).**
>
> Before submission, we found that DISCO-DANCE and UPSIDE had a similar motivation, and that using UPSIDE as a baseline would significantly improve the credibility for our paper. However, UPSIDE requires a set of specific techniques, which makes re-implementation virtually impossible without an open sourced codebase (there is no open sourced code). It was pointed out by the reviewer of the UPSIDE as a [weakness](https://openreview.net/forum?id=25kzAhUB1lz). Since there were no environments shared by the two papers (the size of 2d mazes of two papers is different), the results in UPSIDE could not be simply reported in the main paper.
>
> However, we agree with all reviewers that the discussion and experimental comparison between DISCO-DANCE and UPSIDE are needed. Therefore, from the beginning of the discussion period, we have been in constant contact with the authors of UPSIDE to re-implement the paper to use it as our baseline. The summary of implementation details we utilized are as follows (except for those commonly used in unsupervised skill discovery):
>
> - Reset discriminator everytime it creates new skills.
> - Initialize a new policy and critic for each new skill, for stable training. (100 skills requires 100 actors, 200 critics (including target critic), and 200 optimizers (for actor and critic))
> - For training discriminator, put 3x more weight on ancestor skills compared than leaf node skills, to maintain the tree structure (otherwise newly added skills might steal the region ancestors already reached).
> - Maintain additional replay buffer for all skills including ancestor skills (different from RL replay buffer), and collect the diffusing part of each skill. Train discriminator only with each skill’s diffusing part.
> - Reset the RL replay buffer after new skills are added. Therefore skill z=8 (Figure 12(b)) only learns from the trajectories from z=7,8, indicating that leaf node skill must be executed with its ancestors (i.e., It never learns on states near <starts>, therefore cannot be executed independently from the beginning).
> - When sampling skills (p(z)) for training, instead of uniform random sampling (1/n for discrete skills), all leaf skills are drawn and trained once in a round-robin manner.
> - Training policies K times more than discriminator (e.g., 1 discriminator update and 10 policy updates).
>
> With these details (and other details in the paper), we also tuned multiple hyperparameters including T (length of each skill), N (how many skills are added for children nodes) and learning rates. However, we were unable to reproduce the results in UPSIDE paper in our 2D Mazes.
>
> Therefore, to compare the performance between DISCO-DANCE and UPSIDE, we decided to experiment DISCO-DANCE on environments in UPSIDE paper and report the scores of DISCO-DANCE (we got the environment configuration from the authors recently). We are currently conducting (1) a pretraining experiments on bottleneck-maze and U-maze, and (2) fine-tuning on goal-reaching downstream task. Note that bottleneck-maze in UPSIDE is different in size (5x bigger) from our 2d-bottleneck maze, and requires more training steps. We will report the results of the experiments after the training is finished. We added the above discussions about differences between DISCO-DANCE and UPSIDE in Appendix G.
>
> (For reviewer 2) UPSIDE also has a random walk stage, but what it does is completely different from DISCO-DANCE. In DISCO-DANCE, random walk process (random walk + finding lowest density region via knn) is utilized to select the guide skill, which is orthogonal to the policy π(a|s,z) learning. In contrast, UPSIDE produce rewards from the random walks. Specifically, UPSIDE (1) performs random walks at each skill’s terminal nodes and (2) updates the discriminator with random walked nodes. It induces the local coverage of each skill (i.e., each skill occupies the random walked states, where other skills cannot come into).

---

### Author Response · Authors · 2022-11-15
**Response to all reviewers**

We deeply appreciate all three reviewers for their thoughtful feedback. R1, R2 and R3 indicate reviewer xuUS, reviewer DqoP, reviewer RjRU, respectively. We have posted the manuscript that takes the reviewer’s valuable feedback into account (all modifications are marked in orange). Several experiments are in progress, and we will upload the revised manuscript or report the results in discussion as soon as the experiments are finished. A summary of the **key changes** are shared below:

**Comparison with UPSIDE (R1, R2, R3)**

To compare DISCO-DANCE with UPSIDE, we added explanation of difference two algorithms in Appendix G. **(In progress)** We will also report the experimental results after the training is completed. Please refer to General response #1.

**Efficient random walk process (R2)**

To avoid sample inefficiency in random walk process, we further present a simple and efficient random walk process that requires much less environmental steps but whose performance is almost similar. We reported the performance of proposed efficient random walk process in DMC (Table 4, Figure 7). Please refer to General response #2.

**Correction in experiemental results**

We have corrected two errors in the paper. First, our submission mistakenly did not include random walk process in total pretraining timesteps. We corrected it in the revised manuscript (Table 1,2,4, Figure 5,6,7,13,14,15). Please refer to General response #2. Second, in finetuning experiments, APS had fewer environment steps compared other algorithms. We re-finetuned APS with same environment steps as other algorithms and revised the scores of APS in the manuscript (Table 4, Figure 6,7).

**Additional experiments and ablation studies (R2, R3)**

We conducted sensitivity analysis with respect to the accuracy threshold (which determines apprentice skills) in Appendix C. We reported the state coverage of Ant mazes (Table2) with additional seeds (3 seeds to 5 seeds).

**Visualization of leaned skills on DMC (R2, R3)**

We added qualitative results (visualization of the learned skills) for DMC cheetah environment in Appendix I.

---

### Decision · Program_Chairs · 2023-01-20

**Decision:**

Reject

**Justification For Why Not Higher Score:**

All reviewers agreed the paper could not be accepted in its current form, mostly due to a missing comparison to UPSIDE (a closely related method) and issues in the evaluation protocol (counting the random walk as part of the environment steps budget).

**Justification For Why Not Lower Score:**

N/A

**Metareview: Summary, Strengths And Weaknesses:**

The authors propose a novel algorithm for unsupervised skill discovery, based on the idea of identifying more promising skills which can guide “new” (or unconverged) skills towards under-explored regions of state space. The method appears effective in alleviating the negative exploration bias which has been observed in unsupervised skill discovery algorithms.

While the authors should be commended for providing many improvements to their paper during the rebuttal phase (visualization of DMC skills, more seeds on AntMaze, analysis of sensitivity of hyper-parameters, high-level comparison to UPSIDE), I still cannot recommend acceptance at this point in time. First and foremost is the missing empirical comparison to UPSIDE as highlighted by all reviewers. Given the similarities between both methods, a proper comparison is paramount. While the lack of open-source implementation is unfortunate, as the authors acknowledged, they can always run their methods on environments from (Kamienny et al., 2021). While I appreciated reading the contents of Appendix G (“Comparison to UPSIDE”), I also strongly believe this content should be included in the main paper and not in an appendix. This represents a large undertaking better left to a resubmission.

Lastly, the omission of the random walk steps from the total environment interaction count is also a red flag. While the solution outlined in “general response 2” seems reasonable, I would be uncomfortable basing my decision on such an ad-hoc experimental pipeline. Again, something that time and a resubmission will hopefully address.